# Frataxin deficiency shifts metabolism to promote reactive microglia via glucose catabolism

Francesca Sciarretta[1,10], Fabio Zaccaria[1,2,10], Andrea Ninni[1,2,10], Veronica Ceci[1,2], Riccardo Turchi[1], Savina Apolloni[1], Martina Milani[1,3], Ilaria Della Valle[1,3], Marta Tiberi[4], Valerio Chiurchiù[4,5], Nadia D'Ambrosi[1], Silvia Pedretti[6], Nico Mitro[6,7] ®, Cinzia Volontè[8,9] ®, Susanna Amadio[9], Katia Aquilano[1] ®, Daniele Lettieri-Barbato[1,10] ®

Immunometabolism investigates the intricate relationship between the immune system and cellular metabolism. This study delves into the consequences of mitochondrial frataxin (FXN) depletion, the primary cause of Friedreich's ataxia (FRDA), a debilitating neurodegenerative condition characterized by impaired coordination and muscle control. By using single-cell RNA sequencing, we have identified distinct cellular clusters within the cerebellum of an FRDA mouse model, emphasizing a significant loss in the homeostatic response of microglial cells lacking FXN. Remarkably, these microglia deficient in FXN display heightened reactive responses to inflammatory stimuli. Furthermore, our metabolomic analyses reveal a shift towards glycolysis and itaconate production in these cells. Remarkably, treatment with butyrate counteracts these immunometabolic changes, triggering an antioxidant response via the itaconate-Nrf2-GSH pathways and suppressing the expression of inflammatory genes. Furthermore, we identify Hcar2 (GPR109A) as a mediator involved in restoring the homeostasis of microglia without FXN. Motor function tests conducted on FRDA mice underscore the neuroprotective attributes of butyrate supplementation, enhancing neuromotor performance. In conclusion, our findings elucidate the role of disrupted homeostatic function in cerebellar microglia in the pathogenesis of FRDA. Moreover, they underscore the potential of butyrate to mitigate inflammatory gene expression, correct metabolic imbalances, and improve neuromotor capabilities in FRDA.

## Introduction

Mutations in the frataxin (FXN) gene play a critical role in the development of Friedreich's ataxia (FRDA), a neurodegenerative disorder characterized by progressive muscle weakness and impaired coordination (Clark et al, 2018). In recent times, the significance of neuroinflammation in the context of neurodegenerative disorders has gained substantial recognition and emerging findings highlight a potential involvement of FXN loss in neuroinflammation (Apolloni et al, 2022). FXN deficiency has been shown to increase the production of pro-inflammatory cytokines, suggesting that FXN might be involved in regulating microglial responses (Hayashi et al, 2014; Shen et al, 2016; Khan et al, 2022). FXN is a mitochondrial protein, playing an essential role in the intricate process of iron–sulfur cluster assembly regulating mitochondrial electron transport chain (ETC) and aconitase activity. Loss of FXN has been suggested to disrupt mitochondrial oxidative capacity and cause mitochondrial ROS production (Al-Mahdawi et al, 2006; Anzovino et al, 2014). Aberrant mitochondrial metabolism and increased glycolytic flux are metabolic hallmarks of inflammatory macrophage/microglia activation (Jha et al, 2015; Sangineto et al, 2023). Although it is now well established that FXN takes center place in mitochondrial metabolism, the consequence of FXN loss in microglia cells are poorly explored yet. It has been reported that microbiota-derived short-chain fatty acid butyrate enhances oxidative metabolism and decouples Krebs cycle from glycolytic flux in immune cells (Bachem et al, 2019). Butyrate had a neuroprotective impact on mouse models of Parkinson's disease, likely due to the downstream regulation of gut microbiota and inhibition of gut–brain axis inflammation (Guo et al, 2023). Butyrate reduces neuroinflammation and microglia activation in several experimental models of disease (Huuskonen et al, 2004; Wenzel et al, 2020; Caetano-Silva et al, 2023). Butyrate has been identified as a high-affinity ligand for the Gi-linked heterotrimeric guanine nucleotide-binding protein–coupled receptor (GPCR) hydroxycarboxylic acid receptor 2 (HCAR2) (Carretta et al, 2021), which is expressed in the brain and has been shown to modulate microglial actions in several neuroinflammatory diseases such as multiple sclerosis, Parkinson's disease, and Alzheimer's

---

[1]Department Biology, University of Rome Tor Vergata, Rome, Italy  [2]PhD Program in Evolutionary Biology and Ecology, University of Rome Tor Vergata, Rome, Italy  [3]PhD Program in Cellular and Molecular Biology, University of Rome Tor Vergata, Rome, Italy  [4]Laboratory of Resolution of Neuroinflammation, IRCCS Santa Lucia Foundation, Rome, Italy  [5]Institute of Translational Pharmacology, IFT-CNR, Rome, Italy  [6]DiSFeB, Dipartimento di Scienze Farmacologiche e Biomolecolari "Rodolfo Paoletti", Università degli Studi di Milano, Milano, Italy  [7]Department of Experimental Oncology, IEO, European Institute of Oncology IRCCS, Milan, Italy  [8]National Research Council, Institute for Systems Analysis and Computer Science "A. Ruberti", Rome, Italy  [9]Santa Lucia Foundation IRCCS, Experimental Neuroscience and Neurological Disease Models, Rome, Italy  [10]IRCCS Fondazione Bietti, Rome, Italy

Correspondence: katia.aquilano@uniroma2.it; daniele.lettieri.barbato@uniroma2.it

---

disease (Offermanns, 2014; Moutinho et al, 2022). Recently, diminished abundance of butyrate-producing bacteria has been demonstrated in a mouse model of FRDA (Turchi et al, 2023). Dietary butyrate supplementation in FRDA mice limited macrophage activation in white adipose tissue and in BMDM, suggesting that this molecule could be also efficient in mitigating neuroinflammation and neurobehavior disability.

Herein we demonstrated the FXN deficiency causes an immunometabolic derangement in microglial cells enhancing glucose catabolism to sustain a strong inflammatory phenotype. This evidence was corroborated in an in vivo model of FRDA. Butyrate effectively restored the immunometabolic defects both in vitro and in vivo improving the neuromotor abilities in the FRDA mouse model.

# Results

## FXN deficiency leads to a loss of homeostatic function in cerebellar microglia of knock-in knock-out (KIKO) mice

Although genetic deficiency of the mitochondrial protein FXN is causative of FRDA-related symptoms, the disease-specific cell types in cerebellum are unknown yet. Herein we used a comparative single-cell RNA sequencing (scRNA-seq) between cerebellum of WT and KIKO mice at the early stage of FRDA disease (6 mo old). ScRNA-seq analysis led to the identification of a total of 4,484 quality control (QC)-positive cells. Based on the expression levels of the most variable genes, we annotated homogeneous and robust cluster of cells from scRNA-seq data, resulting in 12 group of cells such as four cluster of endothelial cells (EC), endo-mural cells (EMC), choroid cells, microglia, T cells, pericytes, fibroblasts, astrocytes, and oligodendrocytes (Fig 1A and B). Next, we compared the numerosity of cell clusters between genotype and we found a contraction of choroid plexus cells, astrocytes, and oligodendrocytes (Fig 1C). Oppositely, increased markers of microglia/macrophage population in cerebellum of KIKO were observed (Figs 1B and S1A). Consistently, the gene ontology (GO) terms for biological processes revealed an enrichment in the inflammatory process and down-regulation of mitochondrial oxidative genes (Fig 1D).

Microglia are the primary innate immune cells of the central nervous system (CNS) that are sentinels participating in the inflammatory and cell clearance response (Norris & Kipnis, 2019). To corroborate the microglia dynamics of KIKO mice, we performed a high dimensional flow cytometry analysis (Fig 2A) and we observed that although the percentage of cerebellar $CD45^{low}CD11b^+$ microglia remained preserved, a higher expression of M1-like markers ($CD86^+$ and MHC-II) and a tendency to a lower expression of M2-like marker CD206 was observed in microglial cells of KIKO compared with WT (Fig 2B). No changes were observed in the percentages of neutrophils ($CD45^+/CD11b^-/CD3^-/NK1.1^+/CD90.2^-$ cells) and B cells ($CD45^+/CD11b^-/BB220^+/Ly6G^+$ cells), whereas a significant increase was detected in T cells ($CD45^+/CD11b^-/BB220^-/Ly6G^-/CD3^+/CD90.2^+$ cells) (Fig S1B). Next, to explore the molecular signatures of macrophage/microglia, cerebella $CD45^+/CD11b^+$ cells were isolated by magnetic cell sorting and their transcriptome was profiled by bulk RNA sequencing. We identified n = 1,275 differentially

expressed genes (DEGs) ($-0.75 > Log_2FC > +0.75$; $P < 0.05$) between cerebella $CD45+/CD11b^+$ of KIKO versus WT mice (Fig 2C). GO terms for biological processes of the top 200 down-regulated genes (orange bars) revealed a reduced fatty acid metabolism in microglia/macrophages of KIKO mice (Fig 2D) leading us to suppose a reduced mitochondrial functionality. In an opposite manner, the top 200 up-regulated genes (green bars) pertained to enhanced reactivity to inflammatory stimuli such as LPS, IL1, and TNFa (Fig 2E). Subsequently, we labeled macrophage/microglia in the cerebellum of KIKO mice with CD11b and analyzed cell morphology using the Sholl technique (Ristanovic et al, 2006). Morphological analysis revealed that $CD11b^+$ cells in KIKO mice exhibited longer ramification lengths compared with WT mice (Fig S1C), suggesting an increased reactivity of microglia to inflammatory stimuli (Ziebell et al, 2015; Vidal-Itriago et al, 2022). Moreover, to assess mitochondrial functionality, $CD45^+/CD11b^+$ cells were isolated from the cerebellum, and mitochondrial membrane potential (ΔΨM) was measured. As expected, a reduced ΔΨM was detected in cerebellar macrophage/microglia of KIKO mice compared with WT mice (Fig S1D).

### Loss of FXN enhances glucose catabolism in microglial cells

To further elucidate the molecular mechanisms underlying the observed inflammatory activation in microglia cells derived from the KIKO model, we established a cell model of FRDA by achieving stable down-regulation of FXN expression in a microglial cell line ($BV2^{FXN-}$). Although $BV2^{FXN-}$ cells exhibited lower inflammation than control cells ($BV2^{SCR}$) under basal conditions, as shown in Fig 3A, the greatest reactivity to inflammatory stimuli was observed when $BV2^{FXN-}$ cells were activated with LPS (Fig 3A). These results suggest that LPS-activated $BV2^{FXN-}$ better phenocopy the inflammatory setting observed in cerebella microglia of KIKO mice.

It has been demonstrated that a sustained inflammatory status of macrophages is characterized by metabolic shift from oxidative phosphorylation to glycolysis. To test if a metabolic rearrangement occurs in FXN down-regulating microglia, we measured glycolysis- and TCA-related metabolites in $BV2^{FXN-}$ cells. Notably, mitochondrial metabolites including citrate and oxaloacetate were reduced, whereas αKG was increased in LPS-activated $BV2^{FXN-}$ cells (Fig 3B). On the contrary, glucose uptake (Fig 3C), accumulation of glycolysis and pentose phosphate shunt metabolites (Fig 3D) as well as lactate production (Fig 3E) were significantly increased. These results were consistent with the glucose avidity of inflammatory immune cells (Soto-Heredero et al, 2020). With the aim to explore if the inflammatory phenotype of $BV2^{FXN-}$ cells was dependent on glycolysis, we inhibited glucose uptake by 2-deoxyglucose (2-DG) and as reported in the Fig S2A, a reduced inflammatory response to LPS was observed.

### Itaconate decreases inflammatory responses in FXN-microglia via the Nrf2 pathway

Itaconate, a mitochondrial metabolite, is produced by macrophages in response to inflammatory stimuli to counteract inflammatory stress (Lampropoulou et al, 2016). To test if the highest inflammatory phenotype of LPS-activated $BV2^{FXN-}$ cells was also

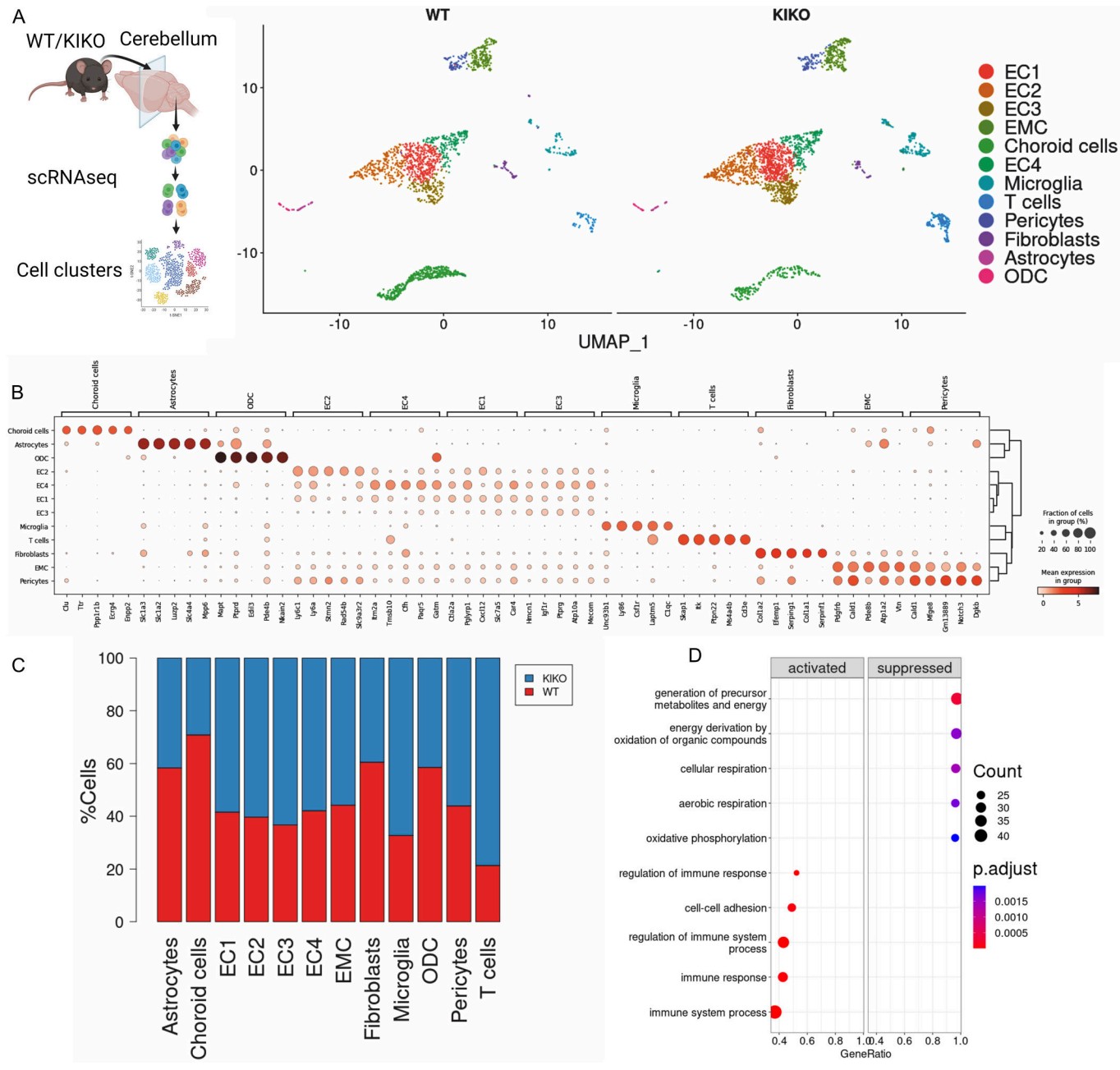

**Figure 1. Cerebellum in knock-in knock-out (KIKO) exhibits an immunometabolic disturbance.**
**(A)** Cell clusters identified by single-cell RNA-seq of total cell fraction (TCF) isolated from cerebellum of WT and KIKO of 6-mo-old mice (TCF: pool from cerebellum of n = 4 mice/group). **(B)** Dot plots reporting gene markers for cell type identified by single-cell RNA-seq of TCF isolated from cerebellum of WT and KIKO of 6-mo-old mice (TCF: pool from cerebellum of n = 4 mice/group). **(C)** Bar plots reporting cell types identified by single-cell RNA-seq of TCF isolated from cerebellum of WT and KIKO of 6-mo-old mice (TCF: pool from cerebellum of n = 4 mice/group). **(D)** Gene Ontology terms for biological processes of differentially expressed genes revelated by single-cell RNA-seq of TCF isolated from cerebellum of WT and KIKO of 6-mo-old mice (TCF: pool from cerebellum of n = 4 mice/group).

associated with itaconate overproduction, we measured its levels, and a significant increase was detected compared to scramble conditions (with or without LPS) (Figs 4A and S2B). Itaconate overproduction observed in BV2$^{FXN-}$ cells was in accordance with the increased expression levels the immune-responsive gene 1 (Irg1) (Fig 4B), the mitochondrial enzyme catalyzing the decarboxylation of cis-aconitate to synthesize itaconate (Lampropoulou

et al, 2016). Remarkably, higher Irg1 expression levels were also detected in cerebellum-derived microglia (Fig 4C). It has been reported that itaconate production following inflammatory stimuli mitigates the inflammatory response by constraining Il1b induction and glycolysis (Lampropoulou et al, 2016). To test if itaconate exerts such\effect in LPS-activated BV2$^{FXN-}$ cells, a cell permeable formulation of soluble itaconate (dimethylitaconate) was added to the

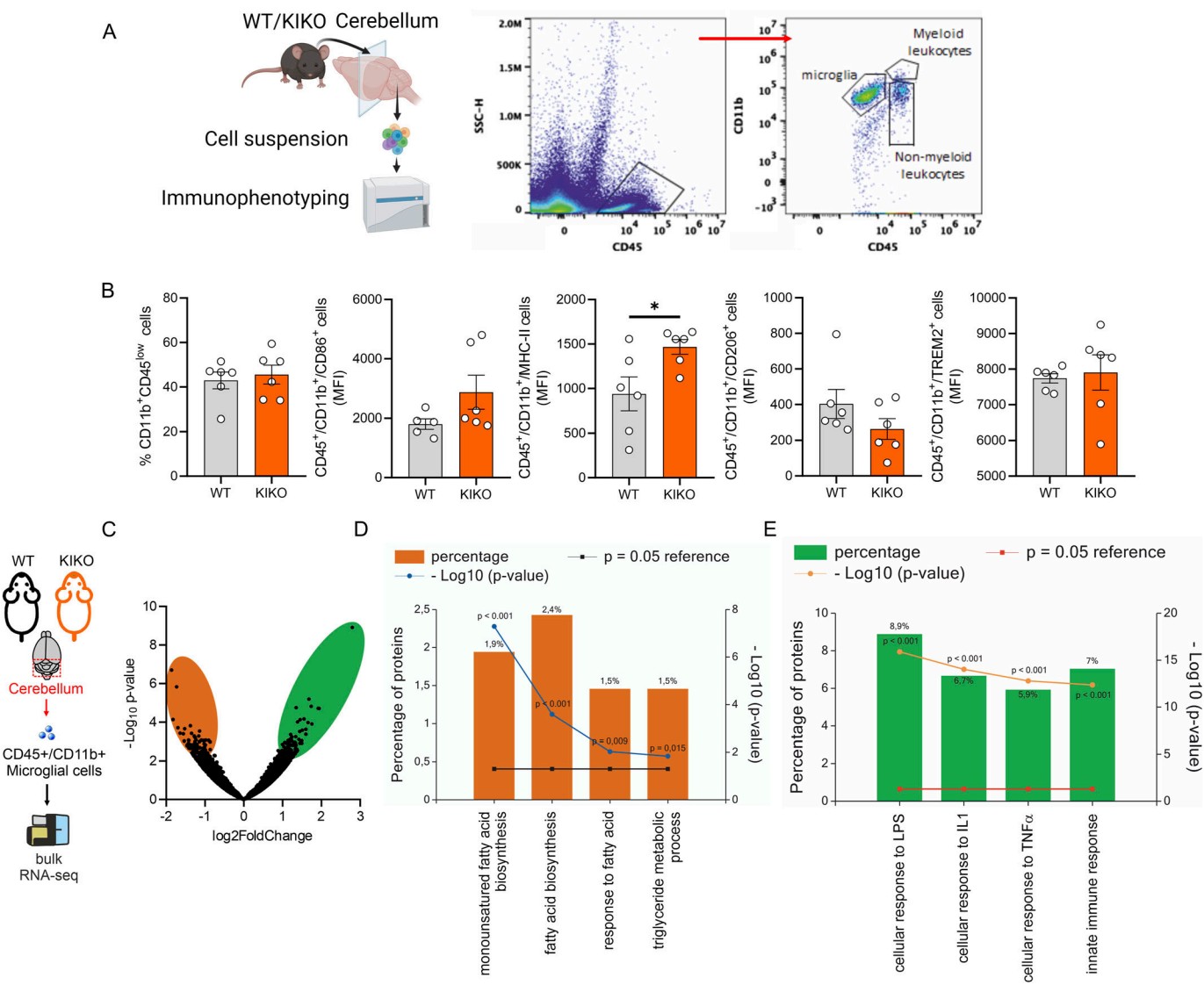

**Figure 2. Microglia-derived from cerebellum of knock-in knock-out (KIKO) mice shows an inflammatory phenotype.**
**(A, B)** High dimensional flow cytometry of pro-inflammatory (CD86 and MHC-II) and anti-inflammatory (CD206 and Trem2) markers in microglial cells (CD4$^{low}$+/CD11b$^+$) isolated from cerebellum of WT and KIKO of 6-mo-old mice (n = 5/6 mice/group). Data were reported as mean ± SD. $t$ test *$P < 0.05$. **(C)** Volcano plot of differentially expressed genes: $-0.75 > Log_2FC > +0.75$; $P < 0.05$ in microglia isolated from 6-mo-old KIKO and WT mice (n = 4 mice/group). **(D, E)** Functional enrichment analysis for biological processes of down-regulated genes ((D) *orange bars*) and up-regulated genes ((E) *green bars*) in microglia isolated from 6-mo-old KIKO and WT mice (n = 4 mice/group).

culture medium. As expected, dimethylitaconate diminished Il1$\beta$, Il6, and Nos2 mRNA levels in BV2$^{FXN-}$ cells (Fig 4D).

### Butyrate reverts the immunometabolic signatures through Itaconate/Nrf2/GSH signaling

Butyrate (BUT) is a ubiquitous short-chain fatty acid principally derived from the enteric microbiome, which showed a neuro-protective role (Li et al, 2016; Lanza et al, 2019). Metabolomic analysis of BUT-treated macrophages revealed a substantial reduction in glycolysis (Flemming, 2019; Schulthess et al, 2019) as well as limited inflammatory response in microglia (Caetano-Silva et al, 2023). By virtue of the recently demonstrated anti-inflammatory

effects of BUT on white adipocytes and BMDM of KIKO mice (Turchi et al, 2023), we asked if butyrate treatment was also effective in counteracting the changes of the immunometabolic profile in LPS-treated BV2$^{FXN-}$. As reported in Fig 5A, BUT reduced glucose uptake and lowered lactate production in LPS-treated BV2$^{FXN-}$ cells, whereas a significant refill in the mitochondrial metabolites such as citrate, oxaloacetate and succinate were observed (Fig 5B). Notably, BUT further increased itaconate levels in BV2$^{FXN-}$ (Fig 5C), leading us to suppose that BUT restored the homeostatic function by the itaconate-driven antioxidant protection. To test this hypothesis, we analyzed Nrf2 protein in BV2$^{FXN-}$ cells treated with butyrate and expectedly an increased nuclear accumulation of Nrf2 was observed (Fig 5D). Nrf2 is the primary transcription factor protecting cells from

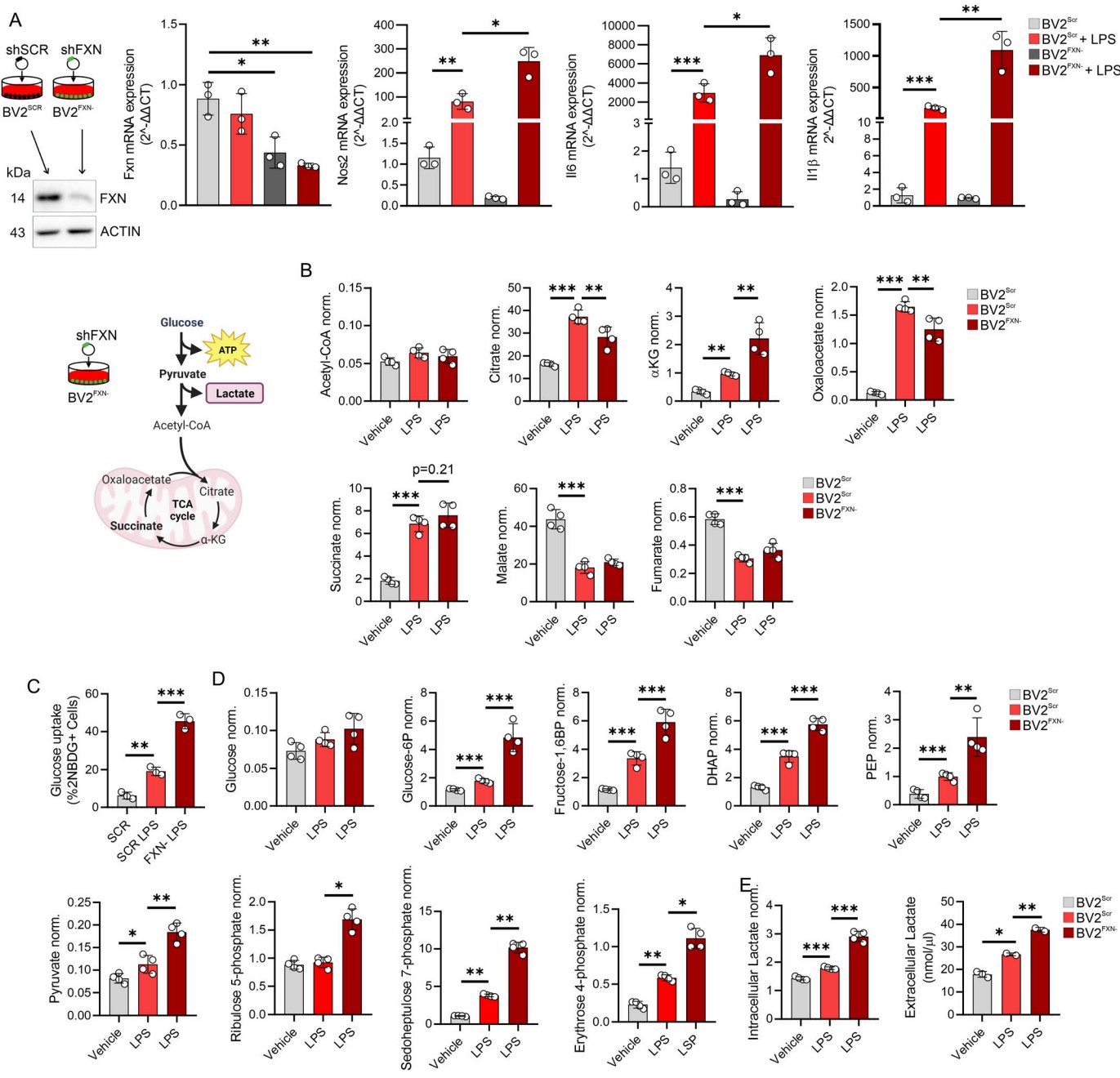

**Figure 3. Decreased frataxin increases glucose catabolism in microglial cells.**
**(A)** BV2 cells were infected with lentiviral particles delivering shRNA against Fxn or Scr sequence and gene expression level of inflammatory genes (Nos2, Il6, and Il1β) were analyzed by qPCR. LPS (500 ng/ml for 16 h) was used to activate BV2 cells. Data were reported as mean ± SD. ANOVA test *P < 0.05; **P < 0.01; ***P < 0.001. **(B)** BV2 were cells infected with lentiviral particles delivering Fxn or Scr sequence and metabolites tracking TCA cycle were measured by LC-MS. LPS (500 ng/ml for 16 h) was used to activate BV2 cells. Data were reported as mean ± SD. ANOVA test **P < 0.01; ***P < 0.001. **(C)** BV2 cells infected with lentiviral particles delivering Fxn or Scr sequence were loaded 2NBDG for 30 min. Glucose uptake calculated as 2-NBDG+ cells by flow cytometry. LPS (500 ng/ml for 16 h) was used to activate BV2 cells. Data were reported as mean ± SD. ANOVA test **P < 0.01; ***P < 0.001. **(D)** BV2 were cells infected with lentiviral particles with shRNA against Fxn or Scr sequence and metabolites tracking glycolysis and pentose phosphate pathway were measured by LC-MS. LPS (500 ng/ml for 16 h) was used to activate BV2 cells. Data were reported as mean ± SD. ANOVA test *P < 0.05; **P < 0.01; ***P < 0.001. **(E)** BV2 were cells infected with lentiviral particles with shRNA against Fxn or Scr sequence and lactate production was measured by LC-MS (intracellular) or spectrophotometer (extracellular). LPS (500 ng/ml for 16 h) was used to activate BV2 cells. Data were reported as mean ± SD. ANOVA test *P < 0.05; **P < 0.01; ***P < 0.001.

oxidative stress by regulating the synthesis of glutathione (GSH) (Harvey et al, 2009). Interestingly, BUT increased GSH levels in BV2^FXN− cells (Fig 5E).

To decipher the molecular mechanism driving the protective effect of butyrate, we analyzed the transcriptomics responses to butyrate in BV2^FXN− (Fig 5F) and KIKO-derived microglial cells

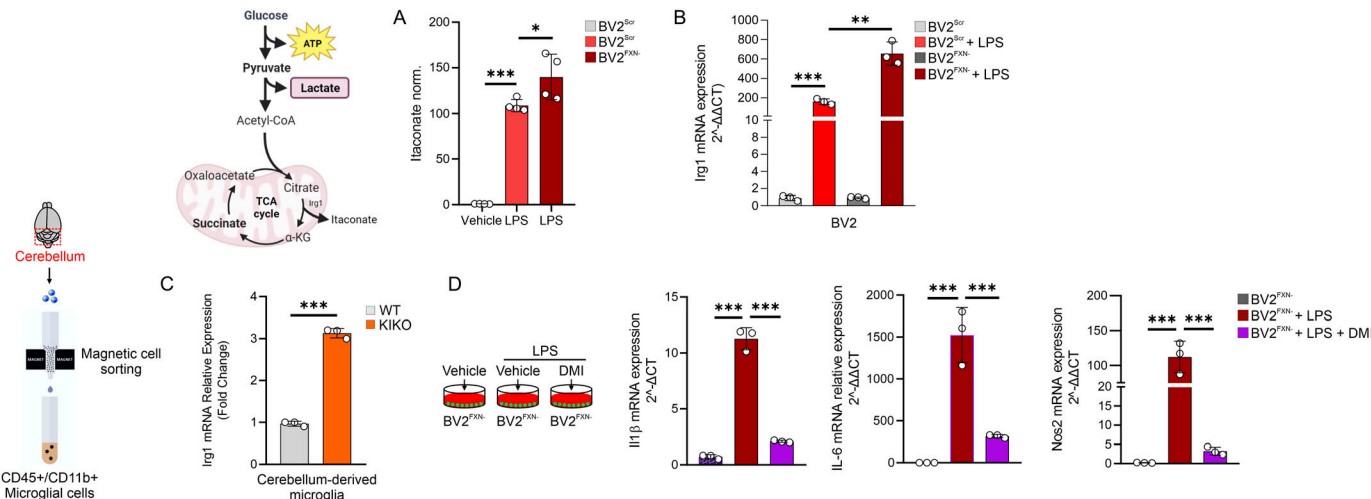

**Figure 4. Itaconate overproduction restrains the inflammatory phenotype in Friedreich's ataxia microglial cells.**
**(A, B)** BV2 cells were infected with lentiviral particles delivering Fxn or Scr sequence and itaconate production (A) and Irg1 mRNA expression (B) were analyzed by LC-MS and qPCR, respectively. LPS (500 ng/ml for 16 h) was used to activate BV2 cells. Data were reported as mean ± SD. ANOVA test *$P < 0.05$; **$P < 0.01$; ***$P < 0.001$. **(C)** Microglial cells were isolated from cerebellum of 6-mo-old knock-in knock-out or WT mice by magnetic cell sorting (CD45[+]/CD11b[+] cells) and Irg1 mRNA expression was analyzed by qPCR. Data were reported as mean ± SD. $t$ test ***$P < 0.001$. **(D)** BV2 cells were infected with lentiviral particles delivering Fxn sequence and the inflammatory gene expression was analyzed by qPCR. LPS (500 ng/ml for 16 h) was used to activate BV2 cells. Dimethylitaconate (100 M) was added 3 h before LPS treatment and maintained throughout the experiment. Data were reported as mean ± SD. ANOVA test ***$P < 0.001$.

(Fig 5G), which showed a high purity as showed by positiveness to CD11b[+] (Fig S2C). The genes that were significantly down-regulated (Log$_2$FC < −1.5) by butyrate were integrated by Venn diagram (Fig 5H) and their functional enrichment analysis suggested that BUT inhibits NfkB signaling pathway in microglia with FXN deficiency (Fig 5I). Consistently, a diminished level of the phospho-active form of Nf-κb was observed in LPS-activated BV2[FXN−] treated with BUT (Fig 5J). To demonstrate the protective effects of BUT in vivo, asymptomatic 4-mo-old KIKO mice were fed with dietary BUT for 16 wk and at the end of dietary treatment, the transcriptome of CD11b[+] microglial cells isolated from cerebellum, was profiled. In accordance with in vitro data, KIKO-derived CD11b[+] microglial cells showed a reduced expression level of inflammatory genes following dietary BUT treatment (Fig S2D).

## Butyrate improves the neuromotor abilities in KIKO mice

Next, we asked if the improvement of the neuroinflammatory status of BUT-treated mice was accompanied by improved neuromotor abilities. To this end, a battery of neuromotor tasks including accelerating rotarod test (Bohlen et al, 2009), pole tests (turning time and climb down time) (Que et al, 2021), and tightrope test (Miquel & Blasco, 1978) were conducted in KIKO mice at the end of dietary treatment. The rotarod test revealed lower neuromotor capacity in KIKO mice compared with the WT mice when the mice ran at maximum RPM (Fig 6A). Nicely, BUT treatment was effective in limiting KIKO falls (Fig 6A). Similar results were observed following pole test, in which KIKO mice showed a highest time to turn completely downward (Tturn) and to descend to the floor (Ttotal) than WT mice (Fig 6B). Although butyrate treatment was effective in improving Tturn (Fig 6B), no improvement was observed in Ttotal (data not shown). Restored neurobehavioral abilities were also

observed at the tightrope test, in which butyrate reduced the higher walking time of KIKO than WT mice (Fig 6C). These results suggest that dietary BUT improves neuromotor abilities through neuro-inflammatory limitation in FRDA mice.

## Hcar2 mediates the anti-inflammatory effects of butyrate

BUT interacts with several G-protein–coupled receptors including GPR109A (encoded by *Hcar2* gene), GPR43 (encoded by *Ffar2* gene) and GPR41 (encoded by *Ffar3* gene) leading to activation of anti-inflammatory signaling cascades (Parada Venegas et al, 2019; Deleu et al, 2021). According to what previously reported (Moutinho et al, 2022), our scRNAseq data revealed that, among these receptors, Hcar2 was expressed at the highest values in microglia (Fig 7A). Interestingly, the Hcar2 expression was higher in KIKO than WT mice (Fig 7B), suggesting an increased sensitivity to BUT in FRDA mice. Remarkably, Hcar2 levels were increased in primary microglia isolated from cerebellum of KIKO mice (Fig 7C), and BUT treatment was effective in restraining its up-regulation (Fig 7C). In line with these findings, BUT limited the expression levels of Hcar2 in LPS-activated BV2[FXN−] cells (Fig 7D). To investigate if Hcar2 mediates the anti-inflammatory effects of BUT, we down-regulated Hcar2 by RNA interference in BV2[FXN−] and the inflammatory genes expression was analyzed following LPS stimulation and BUT pretreatment. Consistent with our hypothesis, BUT was ineffective in limiting inflammatory response in Hcar2 down-regulating cells (Fig 7E).

# Discussion

FXN deficiency caused excess microglial DNA damage and inflammation in murine model of FRDA (Shen et al, 2016). Remarkably,

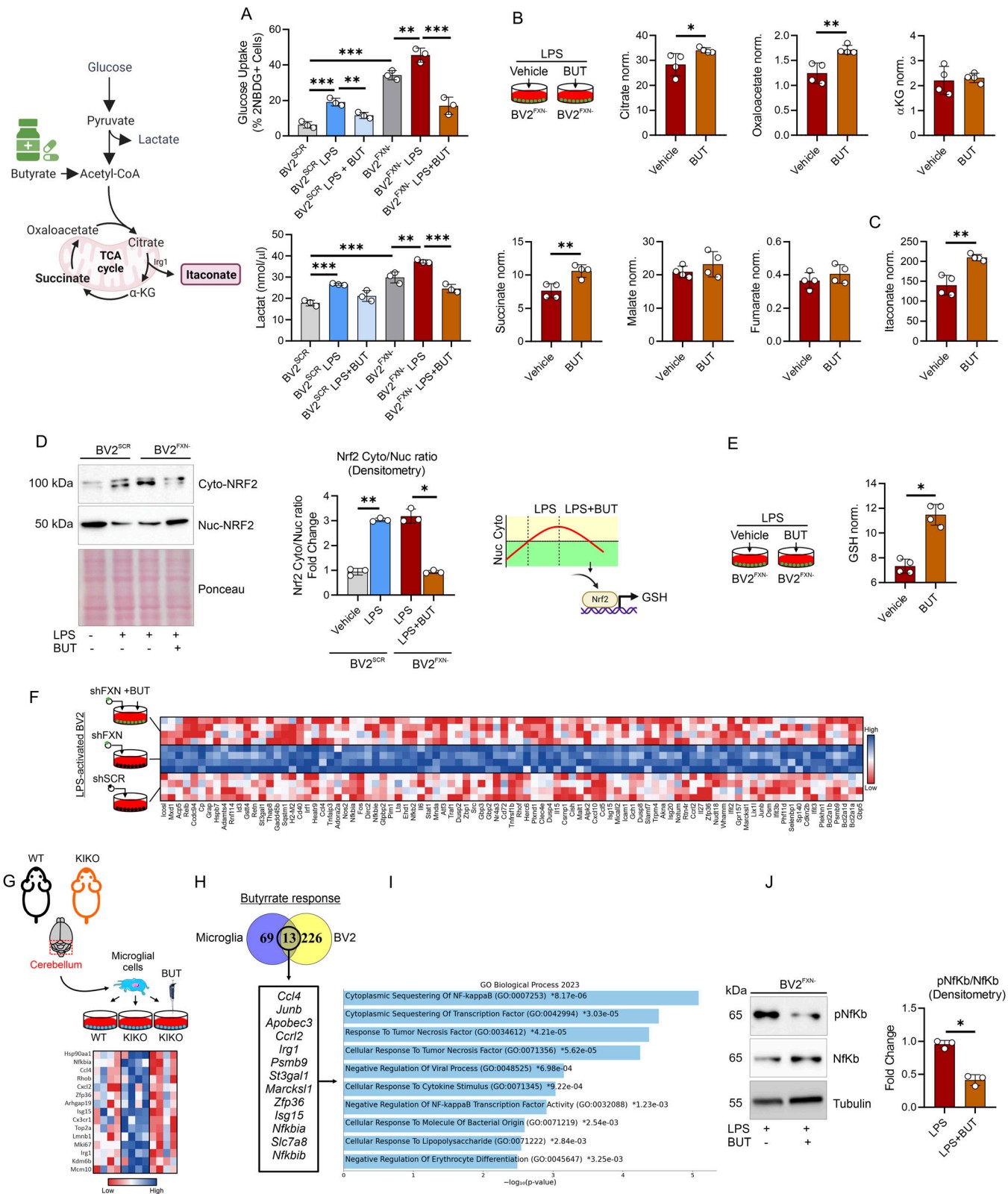

**Figure 5. Butyrate rewires the immunometabolism of microglia with down-regulated frataxin (FXN).**
**(A)** BV2 cells were infected with lentiviral particles delivering Fxn or Scr sequence and glucose uptake (upper panel) and lactate production (bottom panel) were measured by flow cytometry and spectrofluorometer, respectively. LPS (500 ng/ml for 16 h) was used to activate BV2 cells. Sodium butyrate (BUT, 500 μM) was added 3 h before LPS treatment and maintained throughout the experiment. Data were reported as mean ± SD. ANOVA test **$P < 0.01$; ***$P < 0.001$. **(B, C)** BV2 cells were infected with

the transcriptional profile of PBMC isolated from FRDA patients, revealed a strong enrichment for an inflammatory innate immune response (Nachun et al, 2018). Although high inflammatory susceptibility was described in FRDA mice and human (Shen et al, 2016; Khan et al, 2022; Turchi et al, 2023), the mechanisms underlying this condition remain unexplored. Herein we demonstrated that the loss of FXN forces glycolytic catabolism promoting reactive microglia to inflammatory stimuli. FXN is a mitochondrial protein and its dysfunction causes mitochondrial failure, thus recruiting glycolysis as the main source of ATP (O'Neill et al, 2016). This is consistent with the increased glycolysis flux occurring in LPS-treated macrophages and microglial cells (Bernier et al, 2020). Krebs cycle breaks were also described in M1 macrophages and microglia, which cause an overproduction of itaconic acid (Lampropoulou et al, 2016). This mitochondrial metabolite has been shown to participate in the inflammatory response restraining IL1$\beta$ production and glycolysis. Itaconate and its derivatives showed anti-inflammatory effects in preclinical models of sepsis, viral infections, psoriasis, gout, ischemia/reperfusion injury, and pulmonary fibrosis, pointing to possible itaconate-based therapeutics for a range of inflammatory diseases (Peace & O'Neill, 2022). Consistently, itaconate improved the immunometabolic profile in microglia down-regulating FXN through Nrf2-mediated mechanism, highlighting itaconate as novel therapeutical option to improve FRDA-related inflammatory symptoms. It has been reported that itaconate exerts its anti-inflammatory role by activating Nrf2 (Mills et al, 2018). Nrf2 controls the antioxidant responses counteracting the production of oxidatively damaged molecules through GSH synthesis (Mills et al, 2018). In line with this, Nrf2 is down-regulated in FRDA patients and antioxidant GSH precursors are able to counteract oxidative stress in mouse and cellular FRDA models (La Rosa et al, 2021). Although itaconate production requires aconitase activity, which is an iron–sulfur enzyme that may be affected by FXN deficiency, our findings indicate an increase in itaconate production in the KIKO mouse model. We propose several hypotheses that could explain this apparent paradox. It is possible that residual aconitase activity, albeit reduced, is sufficient to support a certain level of itaconate production. The severity of FXN deficiency's impact on aconitase could vary, allowing for some degree of itaconate synthesis to occur. Alternatively, given the role of itaconate as an immunometabolite with anti-inflammatory properties, its increased production might reflect a cellular response to mitochondrial dysfunction and oxidative stress. This

adaptive response could promote the up-regulation of itaconate synthesis as a protective mechanism.

Mounting evidence reports that gut microbiota releases immunomodulatory molecules and counteracts neuroinflammatory conditions (Sampson et al, 2016; Abdel-Haq et al, 2019; Mou et al, 2022; Richards et al, 2022). To this end, targeting gut microbiota has been proposed to alleviate neuroinflammation. Recent meta-genomics profiling revealed that gut microbiota of KIKO mice shows a decrement of butyrate-producing bacteria and dietary butyrate supplementation improves adipose tissue inflammation (Turchi et al, 2023). Dietary butyrate ameliorates microglia-mediated neuroinflammation in several inflammatory mouse models (Jiang et al, 2021; Wei et al, 2023) and improves cognitive decline following neuroinflammatory neurotoxin injection (Ge et al, 2023). In accordance with these data, KIKO mice treated with butyrate show reduced neuroinflammation and improvement of neurobehavioral abilities. In microglial cells down-regulating FXN, we observed that butyrate improves the immunometabolic profile via itaconate/Nrf2/GSH pathway. Butyrate shows a strong chemical similarity to $\beta$-hydroxybutyrate, a ketone body increased in a FRDA mouse model (Dong et al, 2022). However, comparative analyses revealed that butyrate exerts higher impact in terms of induction of the mitochondrial antioxidant genes and inhibition of pro-inflammatory genes (Chriett et al, 2019). It has been suggested that the Nrf2-mediated antioxidant responses induced by butyrate are mediated by Hcar2 (also called as GPR109A) (Guo et al, 2020), which is strongly expressed by CD11b microglial cells (Moutinho et al, 2022). Activation of Hcar2 regulates microglial responses to alleviate neurodegeneration in LPS-induced in vivo and in vitro models (He et al, 2023). In line with this, Hcar2 down-regulation restrained the butyrate-mediated anti-inflammatory responses in FXN-deficient microglia.

The current study provides compelling evidence that the loss of FXN is associated with a disruption in mitochondrial activity, rendering microglial cells highly reactive to inflammatory stimuli. Furthermore, our research indicates that itaconate plays a pivotal role in mitigating this inflammatory cascade through a Nrf2-mediated mechanism. Whereas the protective properties of butyrate have been extensively documented, our study showcases its remarkable ability to ameliorate the neuroinflammatory phenotype through Hcar2-mediated itaconate/Nrf2/GSH signaling pathway. Remarkably, dietary supplementation of butyrate also demonstrated efficacy in enhancing neuromotor

lentiviral particles delivering Fxn sequence and metabolites tracking TCA cycle (B) and itaconate (C) were measured by LC-MS. LPS (500 ng/ml for 16 h) was used to activate BV2 cells. Sodium butyrate (BUT, 500 $\mu$M) was added 3 h before LPS treatment and maintained throughout the experiment. Data were reported as mean ± SD. $t$ test *$P < 0.05$; **$P < 0.01$. **(D)** BV2 cells were infected with lentiviral particles delivering Fxn or Scr sequence and cytosolic/nuclear fractions of NRF2 were analyzed by Western blot (*left panel*). Densitometry was calculated as Cyto-NRF2/Nuc-NRF2 (*right panel*). LPS (500 ng/ml for 16 h) was used to activate BV2 cells. Sodium butyrate (BUT, 500 $\mu$M) was added 3 h before LPS treatment and maintained throughout the experiment. Ponceau staining was used as loading control. **(E)** BV2 cells were infected with lentiviral particles delivering Fxn sequence and GSH and GSSG levels were measured by LC-MS. LPS (500 ng/ml for 16 h) was used to activate BV2 cells. Sodium butyrate (BUT, 500 $\mu$M) was added 3 h before LPS treatment and maintained throughout the experiment. Data were reported as mean ± SD. $t$ test *$P < 0.05$. **(F)** Heat map of differentially expressed genes ($P < 0.05$) in BV2 cells infected with lentiviral particles delivering Fxn or scramble (Scr) sequence. LPS (500 ng/ml for 16 h) was used to activate BV2 cells. Sodium butyrate (BUT, 500 $\mu$M) was added 3 h before LPS treatment and maintained throughout the experiment. **(G)** Heat map of differentially expressed genes ($P < 0.05$) in microglia isolated from the cerebellum of WT and knock-in knock-out mice. Sodium butyrate (BUT, 500 $\mu$M) was added to the culture medium for 16 h. **(H, I)** Venn diagram of butyrate-responsive genes in LPS-stimulated BV2 and microglia isolated from knock-in knock-out mice (H) and the functional enrichment analysis of the overlapping genes was analyzed by EnrichR (I). **(J)** BV2 cells were infected with lentiviral particles delivering Fxn sequence and pospho-active and basal forms of NfKb were analyzed by Western blot (*left panel*). Densitometry was calculated as pNfKb/NfKb (*right panel*). Tubulin was used as loading control. LPS (500 ng/ml for 16 h) was used to activate BV2 cells. Sodium butyrate (BUT, 500 $\mu$M) was added 3 h before LPS treatment and maintained throughout the experiment.

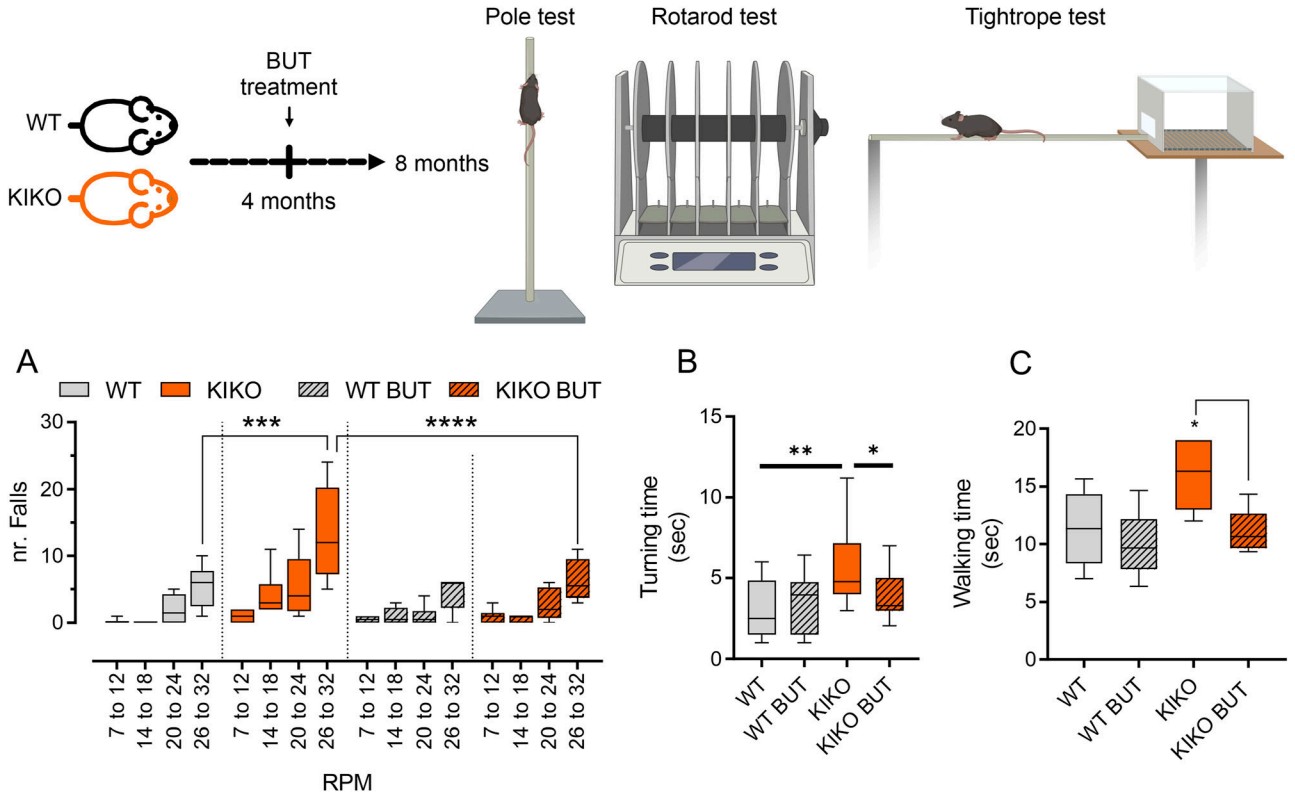

**Figure 6. Butyrate supplementation enhances neuromotor performance in knock-in knock-out mice.**
Male WT and knock-in knock-out mice, aged 4 mo, were either maintained on a standard diet or supplemented with butyrate (BUT) for a duration of 16 wk, until they reached 8 mo of age. **(A)** Rotarod test performance, expressed by the number of falls, across various speeds. **(B)** Duration taken for the mice to turn during pole test atop the pole. **(C)** Time of walking during the tightrope test. Data are presented as mean ± SD. ANOVA *P < 0.05, **P < 0.01, ***P < 0.001, ****P < 0.0001 (n = 6 mice/group).

function in a FRDA mouse model. These findings suggest that butyrate holds significant promise as a readily accessible and safe therapeutic option for alleviating FRDA neurological symptoms.

### Limitation of the study

In our investigations involving BV2 cells, we used a single shRNA construct. Although this approach allowed us to assess FXN down-regulation, it is essential to acknowledge the potential for off-target effects associated with shRNA techniques. Thus, a notable limitation of our study lies in the absence of additional shRNA sequences to validate the observed FXN down-regulation in BV2 cells. Moreover, an important aspect hindering the comprehensive understanding of itaconate accumulation in FXN-deficient BV2 cells relates to the observed imbalance in metabolites, notably the low levels of citrate juxtaposed with elevated levels of aKG. This metabolic perturbation complicates the interpretation of itaconate accumulation dynamics. To address this challenge and enhance our understanding, future studies could use metabolic flux analysis techniques alongside the assessment of itaconate derivatives. Such approaches would provide deeper insights into the metabolic alterations underlying itaconate accumulation in FXN-deficient BV2 cells.

## Materials and Methods

### Mice and treatments

#### WT and KIKO mice
Mouse experimentation was carried out in strict accordance with established standards for the humane care of animals, following approval by the relevant local authorities, including the Institutional Animal Care and Use Committee at Tor Vergata University, and national regulatory bodies (Ministry of Health, licenses no. 324/218-PR and no. 210/202-PR). Both female and male mice were housed in controlled conditions, with a temperature of 21.0°C and a relative humidity of 55.0% ± 5.0%, all while adhering to a 12-h light/12-h dark cycle (lights on at 6:00 AM, lights off at 6:00 PM). They were provided with unrestricted access to food and water, and all experimental procedures were conducted in accordance with institutional safety protocols. The female and male KIKO mice were obtained from Jackson Laboratories (#012329), and their female and male littermate C57BL/6 counterparts (WT) were used as control subjects. During testing, the researchers were unaware of the genotypes to ensure unbiased results.

The supplementation of butyrate in male mice was performed as previously reported (Turchi et al, 2023). Sodium butyrate was incorporated into their food pellets (at a rate of 5 g per kg per day, consistent with their regular daily caloric intake) starting at 4 mo of age. This age was selected as it precedes the onset of metabolic

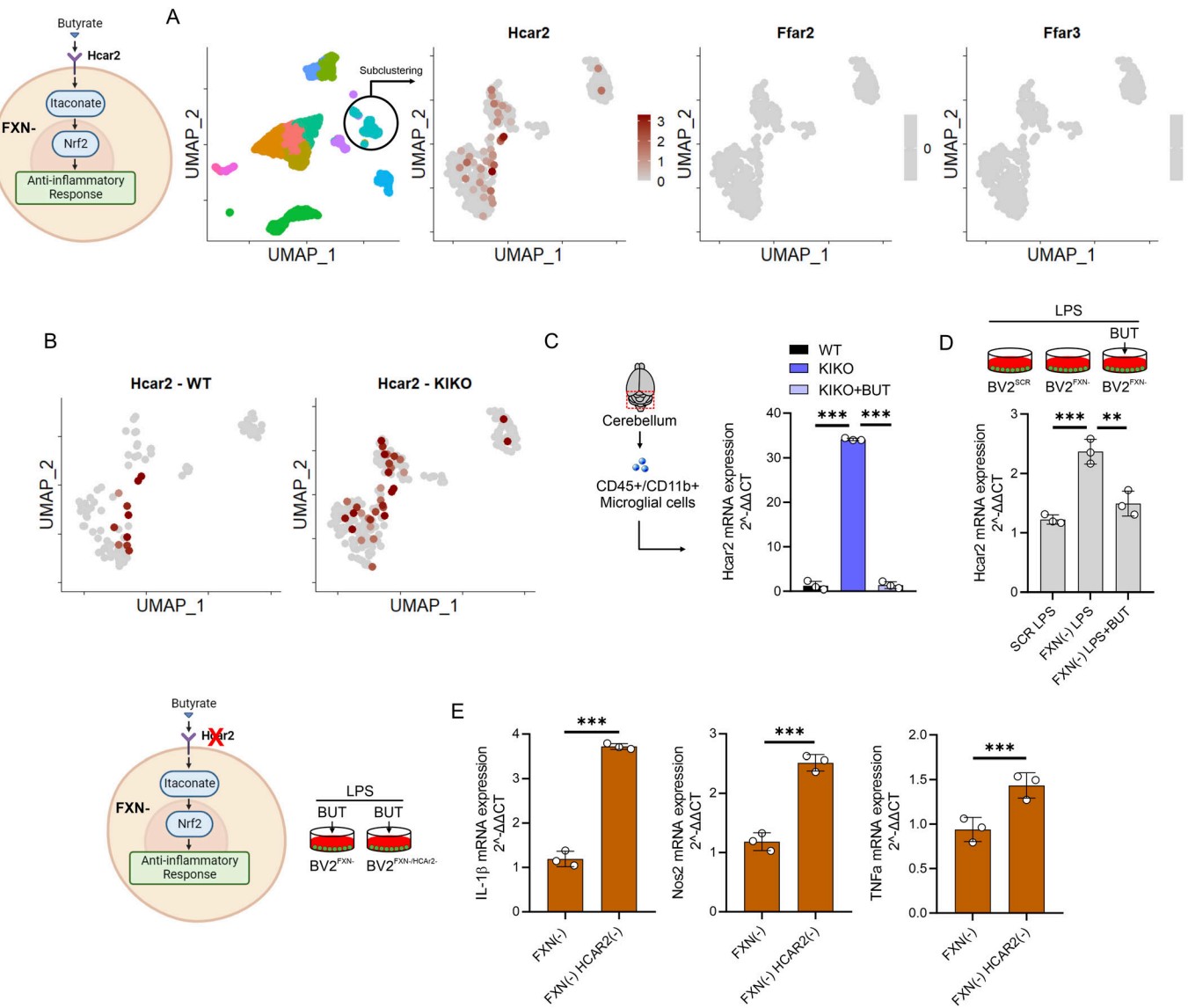

**Figure 7. Hcar2 mediates the butyrate responses in the Friedreich's ataxia microglia.**
**(A)** Cerebellar microglial cells analyzed by scRNA-seq were subclustered and Hcar2, Ffar2, and Ffar3 expression levels were analyzed (pool of n = 4 mice/group).
**(B)** Hcar2 expression levels in cerebellar microglia of 6-mo-old WT and knock-in knock-out (KIKO) mice (pool of n = 4 mice/group). **(C)** Microglia were isolated from cerebellum of WT, KIKO, or KIKO mice fed with butyrate and Hcar2 expression level was measured by qPCR (n = 3 mice/group). Data were reported as mean ± SD. ANOVA test ***$P < 0.00$. **(D)** BV2 cells were infected with lentiviral particles delivering Fxn or Scr sequence and mRNA expression of Hcar2 was measured by qPCR. LPS (500 ng/ml for 16 h) was used to activate BV2 cells. Sodium butyrate (BUT, 500 μM) was added 3 h before LPS treatment and maintained throughout the experiment. Data were reported as mean ± SD. ANOVA test **$P < 0.01$; ***$P < 0.001$. **(E)** BV2 FXN⁻ cells were infected with lentiviral particles delivering scramble (Scr) or *Hcar2* RNAi sequence. Hcar2 expression levels and inflammatory cytokines expression level was measured by qPCR. Data were reported as mean ± SD. ANOVA test **$P < 0.01$; ***$P < 0.001$.

changes and continued until the mice reached 8 mo of age, which corresponds to a 16-wk treatment period. This timeline was chosen because it coincides with the point at which mice typically begin to display metabolic alterations and weight gain. At 8 mo of age, the mice were euthanized by cervical dislocation and cerebellum was immediately processed or stored at –80°C for subsequent analysis.

### Neurobehavioral tests

Before rotarod testing, mice were trained for 1 d on the rotarod (cat. n. 47600; Ugo Basile s.r.l.) at a constant speed of 7 rpm over 1 min, repeated four times. On the day of testing, each mouse was placed on a stationary rod which was then accelerated from 7 to 32 rpm over 5 min. The latency to fall was recorded. This was performed over three trials with 60-min inter-trial intervals.

Before pole testing, mice were acclimated to a wooden pole measuring 30 cm × 1 cm. For the turning time assessment, each mouse was placed head upwards at the top of the pole. The time taken for the mouse to turn 180° downward was recorded. The descent time, from turning to reaching the base of the pole, was subsequently documented.

For tightrope test, a rope measuring 60 cm in length and 1 cm in diameter was securely stretched between two platforms. Each mouse was placed at the center of the rope, and the time taken to reach either platform was noted. This procedure was repeated over three trials with 30-min intervals between trials.

## Cells and treatments

### Primary microglia isolation and BV2 cell line

Primary microglia from the cerebellum were isolated following a previously described method (Apolloni et al, 2013). In brief, mice at 5–6 d of age (p6) were euthanized, and the meninges were carefully removed. The cerebellum was then finely chopped and subjected to digestion using 0.01% trypsin and 10 $\mu$g/ml DNaseI. After dissociation and filtration through 70-$\mu$m filters, cells were suspended in DMEM/F-12 medium supplemented with GlutaMAX (Gibco, Invitrogen). This medium was further supplemented with 10% FBS, 100 U/ml of gentamicin, and 100 $\mu$g/ml of streptomycin/penicillin. The cells were plated at a density of 62,500 cells per cm². After ~15 d, a gentle trypsinization was performed using DMEM/F-12 without FBS (0.08% trypsin in DMEM/F-12 without FBS) for 40 min at 37°C to eliminate non-microglial cells. The resulting adherent microglial cells were then cultured in a mixture of glial cell-conditioned medium (50%) at 37°C in an atmosphere containing 5% $CO_2$ for 48 h before use.

To isolate cerebella microglia by magnetic cell sorting, cerebellum homogenate as resuspended in 500 ml of magnetic bead buffer (MBB) consisting of PBS without calcium and magnesium, 0.5% wt/vol BSA, and 2 mM EDTA. The cell suspension was then filtered through a 30-mm pre-separation filter (Miltenyi) following three filter washes to remove any large particles and debris. The resulting cell suspension was then separated at 300$g$ for 5 min at 4°C and resuspended in MBB along with antiCD45 magnetic beads-conjugated antibody (Miltenyi). The cell suspension was incubated for 15 min at 4°C, then diluted with 2 ml of MBB and centrifuged. The resulting cell pellet was resuspended in 500 ml of MBB, applied onto hydrated MS-columns (Miltenyi), washed three times with 500 ml of MBB, and collected with 1 ml of MBB through piston elution. CD45[+] cells resuspended in MBB along with anti-CD11b magnetic beads-conjugated antibody (Miltenyi). The cell suspension was incubated for 15 min at 4°C, then diluted with MBB and centrifuged. The resulting cell pellet was resuspended in 500 ml of MBB, applied onto hydrated MS-columns (Miltenyi), washed three times with 500 ml of MBB, and collected with 1 ml of MBB through piston elution to obtain CD45[+]/CD11b[+] cells.

Murine BV2 cell line (ATCC) was cultured in DMEM supplemented with 10% FBS and 1% P/S (Life Technologies) and 1% non-essential amino acids (Euroclone). All cells were maintained at 37°C in a humidified incubator containing 5% $CO_2$. For gene silencing, BV2 cells were seeded 20,000 cell/well. 24 h after plating, BV2 cells were infected with 25 MOI of FXN shRNA or scramble shRNA (Origene) for a total of 500,000 viral particles/well. The Fxn mouse shRNA lentiviral particle (Locus ID 14297) was generated using the pGFP-C-shLenti vector (TR30023). To facilitate viral particle entry in the cells 2 $\mu$g/ml polybrene (Sigma-Aldrich), we added to the culture medium. BV2 cells were treated with 500 ng/ml LPS for 16 h. Sodium butyrate (BUT, 500 $\mu$M) was added 3 h before LPS treatment and maintained throughout the experiment. The sodium butyrate concentration was selected based on dose-response experiments conducted on primary adipocytes or bone marrow-derive macrophages stimulated with LPS (500 ng/ml, 16 h). These experiments demonstrated the anti-inflammatory action of the 500 mM concentration while preserving cell viability (Turchi et al, 2023).

### Generation of cerebellar single-cell suspensions

Mouse cerebella were dissected in ice-cold PBS. Tissues from four animals were pooled and then cut into pieces smaller than 1 mm in size. These pieces were then transferred to ice-cold MACS Tissue Storage Solution (Miltenyi Biotec). For cell dissociation, the storage solution was replaced with RPMI 1640 Medium (Thermo Fisher Scientific). The tissue was pelleted and digested with 500 ml of pre-warmed Accumax (Innovative Cell Technologies) in a 1.5-ml tube at 37°C for 5 min. After digestion, the tissue pieces were dissociated by gentle trituration with a wide-bore pipet tip. The cell suspension was then filtered through a 30-mm pre-separation filter (Miltenyi) following three filter washes to remove any large particles and debris. Any remaining cell clumps unable to pass through the filter were subjected to another round of digestion and filtering to maximize the yield of single cells. The cell suspension was collected into 1.5 ml of ice-cold Resuspension Buffer (Lebovitz L15 medium with 2% FBS, 25 mM HEPES, 2 mM EDTA). After dissociation, cells were stained with Trypan blue, counted, and cryopreserved in 90% FBS/10% DMSO freezing medium. Single-cell suspensions with more than 85% viability were then used to generate single-cell cDNA libraries.

### Single-cell RNA sequencing

Single-cell suspensions were prepared for scRNA-seq immediately after thawing, using the Chromium Single-Cell Reagent Kit from 10x Genomics, following the manufacturer's protocol. After cell capture and lysis, cDNA was synthesized for each group of captured cells and underwent 12 cycles of amplification. The amplified cDNA from each channel of the Chromium system was used to construct an Illumina sequencing library, which was sequenced using the NovaSeq 6000, resulting in ~300 million reads per library with a 2 × 50 read length. Raw reads were aligned to the *Mus musculus* (mm10) reference genome, and cells were identified using Cell-Ranger count v.7.1.0. Individual samples were combined to create a merged digital expression matrix. The barcodes, features, and matrix files generated by the CellRanger software were used as input for the R program Seurat v4.4.0 (Satija et al, 2015). KIKO sample: the raw data returned 32,285 genes (features) and 3,353 cells (barcodes), after the quality control and filtering steps were retained only cells with more than 500 features, gene counts greater than 1,000, and mitochondrial content less than 10%. Outliers, defined as cells with more than 10,000 features and counts exceeding 20,000, were removed, resulting in a final subset composed of 2,358 cells. WT sample: 2,854 cells were filtered by the same approach used for the KIKO sample, resulting in a final WT subset composed of 2,442 cells.

Expression levels were normalized using logarithmic transformation. The most variable genes (2,000 features) were selected, and their expressions were scaled across all cells. The dimensionality of the dataset was assessed through Principal Component Analysis,

and the first 20 principal components were used to create a UMAP reduction. Clustering was performed with a resolution parameter set to 0.5. DEGs within each cluster were identified using the Wilcoxon rank sum test. Enrichment analysis was carried out using ClusterProfileR v4.4.4, PMID: 22455463, which identified the top-5 activated and top-5 suppressed Gene Ontology Biological Processes terms using the gseGO function. Subsequently, the Microglia cluster was isolated from the main dataset and subjected to full reprocessing. Single-cell plots were generated using the GGPlot2 package v3.4.4.

## Bulk RNA sequencing

Total RNAs were extracted from cells and tissues using the MiniPrep kit from ZYMO RESEARCH, in adherence to the manufacturer's instructions. The quantification of total RNA was performed using the Qubit 4.0 Fluorimetric Assay from Thermo Fisher Scientific. Libraries were constructed from 50 ng of total RNA through the NEGEDIA Digital mRNA-seq research grade sequencing service provided by Next Generation Diagnostic srl. This service encompassed library preparation, quality assessment, and sequencing on an Illumina NovaSeq 6000 system using a single-end, 75-cycle strategy. The raw data underwent analysis with FastQC v0.12.0 and were subsequently subjected to quality filtering and trimming by Trimmomatic v0.39 (Bolger et al, 2014), using a Q30 threshold for both leading and trailing ends, with a minimum length of 15 nucleotides. The resultant reads were aligned to the reference genome (mm10) using HISAT2 v2.2.1 (Kim et al, 2019). Quantification of gene expression was accomplished using the featureCounts tool (Liao et al, 2014). Subsequently, the DESeq2 package v1.40.2 (Love et al, 2014) was used to calculate DEGs and normalize the expression count matrix. Functional enrichment analysis was carried out using EnrichR or FunRich 3.0 with the Biological Processes Gene Ontology (GO) database.

## Immunophenotyping by flow cytometry

Cerebellum was immunophenotyped by high dimensional flow cytometry using a panel containing markers to identify cell types and to assess activation states. The use of these markers allowed us to exclude all cells of no interest based on physical parameters (side and forward scatter) and to gate on specific cells of interest. In particular, cerebellum of WT and KIKO mice was dissociated to single-cell suspension using adult brain dissociation kit from MACS Technology and using GentleMACS (Miltenyi Biotec), according to the manufacturer's protocol. Cells were first gated on CD45$^+$ cells and then on CD45$^{low}$CD11b$^+$ to identify microglial cells and to exclude infiltrated macrophages (CD45$^{high}$CD11b$^+$) and non-myeloid leukocytes (CD45$^{high}$CD11-b$^{low}$). Microglia were further stained for the expression of M1 (anti-CD86 and anti-MHC-II) or M2 (anti-CD206 and anti-Trem2) markers. Samples were acquired on a 13-color Cytoflex (Beckman Coulter) and for each analysis, at least $0.5 \times 10^6$ live cells were acquired by gating on aqua Live/Dead negative cells (Sciarretta et al, 2023).

## Sholl analysis

Confocal assay analysis was performed using the ImageJ tool (v1.54f). From each scan, two layers combining nuclei (cyan channel) and microglia (blue) were selected and merged. Next, the microglia channel was analyzed after a grey-scale conversion. A total of 10 cells per sample were selected for further analysis, among these four cells per sample were selected for visualization purposes. Each microglia cell was studied by the ImageJ plugin "Neuroanatomy": more in detail, the "Simple Neurite Tracker" (SNT—v4.2.1) was used to analyze microglia cell structures. The original scan images were cropped to study single microglia cells. Each resulting image was segmented and simplified by the "skeletonize" function. The major branches of each selected cell were measured by the SNT tool. The main ramifications were also evaluated. For each cell studied, each major branch and/or ramification was measured, then the mean length of every branch was calculated and subsequently the total branch length of every cell was summed and averaged across the same sample cells. Then, the occupancy radius of each microglia cell was calculated using the Sholl tool of the Neuroanatomy plugin. The resulting data were analyzed and plotted by GraphPad Prism (v9.1.0).

## Targeted metabolomics

All data were acquired on a Triple Quad API3500 (AB Sciex) with an HPLC system ExionLC AC System (AB Sciex). For targeted metabolomic analysis, cells were extracted using tissue lyser for 30 s at maximum speed in 250 $\mu$l of ice-cold methanol: water: acetonitrile (55:25:20) containing [U-$^{13}$C$_6$]-glucose 1 ng/$\mu$l and [U-$^{13}$C$_5$]-glutamine 1 ng/$\mu$l as internal standards (Merk Life Science). Lysates were spun at 15,000$g$ for 15 min at 4°C, dried under N$_2$ flow at 40°C, and resuspended in 125 $\mu$l of ice-cold methanol/water 70:30 for subsequent analyses. Amino acids analysis was performed through the previous derivatization. Briefly, 50 $\mu$l of 5% phenyl isothiocyanate in 31.5% ethanol and 31.5% pyridine in water were added to 10 $\mu$l of each sample. Mixtures were then incubated with phenyl isothiocyanate solution for 20 min at RT, dried under N2 flow, and suspended in 100 $\mu$l of 5 mM ammonium acetate in methanol/H$_2$O 1:1. Quantification of different amino acids was performed by using a C18 column (Biocrates) maintained at 50°C. The mobile phases were phase A: 0.2% formic acid in water and phase B: 0.2% formic acid in acetonitrile. The gradient was T$_0$: 100% A, T$_{5.5}$: 5% A, and T$_7$: 100% A with a flow rate of 500 $\mu$l/min. Measurement of energy metabolites and cofactors was performed by using a cyano-phase LUNA column (50 mm × 4.6 mm, 5 $\mu$m; Phenomenex), maintained at 53°C, by a 5-min run in negative ion mode. The mobile phase A was water, whereas phase B was 2 mM ammonium acetate in MeOH, and the gradient was 50% A and 50% B for the whole analysis, with a flow rate of 500 $\mu$l/min. Acylcarnitines quantification was performed using a ZORBAX SB-CN 2.1 × 150 mm, 5 $\mu$m column (Agilent). Samples were analyzed by a 10-min run in positive ion mode. The mobile phases were phase A: 0.2% formic acid in water and phase B: 0.2% formic acid in acetonitrile. The gradient was T$_0$: 100% A, T$_{5.5}$: 5% A, and T$_7$: 100% A with a flow rate of 350 $\mu$l/min. All metabolites analyzed were previously validated by pure standards, and internal standards were used to check

instrument sensitivity. MultiQuant software (version 3.0.3; AB Sciex) was used for data analysis and peak review of chromatograms. Data were normalized on the median of the areas and then used to perform the statistical analysis.

## Lactate production and glucose uptake

Extracellular lactate levels were assessed in the culture medium via an enzyme-based spectrophotometric assay. The procedure involved the collection of cell media, followed by treatment with a 1:2 (vol/vol) solution of 30% trichloroacetic acid to precipitate proteins. Afterward, the resulting mixture was subjected to centrifugation at 14,000$g$ for 20 min at 4°C, and the supernatant was carefully collected. Subsequently, the collected supernatant was incubated for 30 min at 37°C with a reaction buffer containing glycine, hydrazine, NAD+, and LDH (lactate dehydrogenase) enzyme. This incubation allowed for the conversion of lactate to pyruvate, while simultaneously reducing NAD+ to NADH. The concentration of NADH, which is stoichiometrically equivalent to the amount of lactate, was then determined at 340 nm using a spectrophotometer.

To monitor glucose uptake, 2-NBDG probes were used according to manufactures protocols. Flow cytometry analyses were performed by 13-color Cytoflex (Beckman Coulter) and the percentage of 2-NBDG–positive cells was calculated by FlowJo software.

## Quantitative PCR (qPCR)

The total RNA was isolated using TRI Reagent (Sigma-Aldrich). Subsequently, 3 mg of RNA was reverse-transcribed with M-MLV (Promega). qPCR was performed in triplicate, using validated qPCR primers confirmed via BLAST searches. The Applied Biosystems Power SYBR Green Master Mix was used, along with the QuantStudio3 Real-Time PCR System (Thermo Fisher Scientific). mRNA levels were normalized to actin mRNA, and the relative mRNA levels were determined using the $2^{-\Delta\Delta Ct}$ method. The primers used for reverse transcription quantitative PCR (RT-qPCR) are as follows:

Fxn:
Forward: 5'-TCTCTTTTGGGGATGGCGTG-3'
Reverse: 5'-GCTTGTTTGGGGTCTGCTTG-3'
Il1b:
Forward: 5'-TGCACCTTTTGACAGTGATG-3'
Reverse: 5'-AAGGTCCACGGGAAAGACAC-3'
Il6:
Forward: 5'-GGATACCACTCCCAACAGA-3'
Reverse: 5'-GCCATTGCACAACTCTTTTCTCA-3'
Rpl8:
Forward: 5'-GGAGCGACACGGCTACATTA-3'
Reverse: 5'-CCGATATTCAGCTGGGCCTT-3'
Nos2:
Forward: 5'-GCCTTCAACACCAAGGTTGTC-3'
Reverse: 5'-ACCACCAGCAGTAGTTGCTC-3'
Hcar2:
Forward: 5'-GAGCAGTTTTGGTTGCGAGG-3'
Reverse: 5'-GGGTGCATCTGGGACTCAAA-3'
Irg1:

Forward: 5'-GCAACATGATGCTCAAGTC-3'
Reverse: 5'-TGCTCCTCCGAATGATACCA-3'
Tnfa:
Forward: 5'-ATGGCCTCCTCATCAGTTC-3'
Reverse: 5'-TTGGTTTGCTACGACGTG-3'

## Immunoblotting

Tissues or cells were lysed in RIPA buffer containing 50 mM Tris–HCl (pH 8.0), 150 mM NaCl, 12 mM deoxycholic acid, 0.5% Nonidet P-40, as well as protease and phosphatase inhibitors. Next, 5 mg of proteins were loaded onto an SDS–PAGE gel and subjected to Western blotting. Nitrocellulose membranes were subsequently incubated with primary antibodies at a 1:1,000 dilution. Following this, the membranes were incubated with the appropriate horseradish peroxidase-conjugated secondary antibodies. Immunoreactive bands were detected using a FluorChem FC3 System (ProteinSimple) after the membranes were incubated with ECL Prime Western blotting Reagent (GE Healthcare). Densitometric analysis of the immunoreactive bands was performed using the FluorChem FC3 analysis software. The antibodies used for immunoblotting as are follows: NRF2 (cod.: SAB4501984; Sigma-Aldrich); pNfKb (cod..; #30335; Cell Signalling); NfKb (cod.: #4764; Cell Signalling); Tubulin (cod.: 10094-1-AP; Proteintech).

## Statistical analysis

The data were presented as the mean ± SD. The specific number of replicates for each dataset is provided in the corresponding figure legend. To evaluate the statistical significance between two groups, a two-tailed unpaired $t$ test was conducted. For comparisons involving three or more groups, an ANOVA was performed, followed by either Dunnett's test (for comparisons relative to controls) or Tukey's test (for multiple comparisons among groups). These statistical analyses were carried out using GraphPad Prism 9 (GraphPad Software Inc.). In all instances, a significance threshold of $P < 0.05$ was set.

# Data Availability

All raw data that support the findings of this study are available from the lead contact upon reasonable request. scRNA-seq and bulk RNA-seq datasets produced in this study are available from gene expression omnibus (GEO) with accession number GSE261655.

# Supplementary Information

# Acknowledgements

This work was supported by Friedreich's Ataxia Research Alliance (FARA)-General Research Grant 2021 to D Lettieri-Barbato. Other supports: Progetto Giovani Ricercatori, Italian Ministry of Health (GR-2018-12367588) to D

Lettieri-Barbato; FARA-General Research Grant 2020 to K Aquilano; Office of the Assistant Secretary of Defense for Health Affairs endorsed by Department of Defense (USA) through the Congressionally Directed Medical Research Programs Award (No. HT9425-23-1-0005) to K Aquilano and D Lettieri-Barbato; National Ataxia Foundation (NAF) (821396[RG]) to N D'Ambrosi; #NEXTGENERATIONEU (NGEU)—Ministry of University and Research (MUR), National Recovery and Resilience Plan (NRRP), project MNESYS (PE0000006), A Multiscale integrated approach to the study of the nervous system in health and disease (DN. 1553 11.10.2022) to D Lettieri-Barbato, K Aquilano, and N D'Ambrosi. Ministry of University and Research (MUR) Progetto Eccellenza (2023–2027) to the Department of Pharmacological and Biomolecular Sciences "Rodolfo Paoletti" (S Pedretti and N Mitro), Università degli Studi di Milano, and partially by the Italian Ministry of Health with Ricerca Corrente and 5 × 1,000 funds to N Mitro.

## Author Contributions

F Sciarretta: data curation, formal analysis, and investigation.
F Zaccaria: data curation and formal analysis.
A Ninni: data curation and formal analysis.
V Ceci: formal analysis.
R Turchi: formal analysis.
S Apolloni: formal analysis.
M Milani: formal analysis and methodology.
I Della Valle: formal analysis and methodology.
M Tiberi: formal analysis.
V Chiurchiù: formal analysis and methodology.
N D'Ambrosi: formal analysis and methodology.
S Pedretti: formal analysis and methodology.
N Mitro: formal analysis and methodology.
C Volontè: formal analysis and methodology.
S Amadio: formal analysis and methodology.
K Aquilano: supervision, validation, and writing—review and editing.
D Lettieri-Barbato: conceptualization, supervision, project administration, and writing—original draft, review, and editing.

## Conflict of Interest Statement

The authors declare that they have no conflict of interest.

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
