## [Reviewer comments · Life Science Alliance]

Life Science Alliance

Frataxin Deficiency Shifts Metabolism to Promote Reactive Microglia via Glucose Catabolism

Francesca Sciarretta, Fabio Zaccaria, Andrea Ninni, Veronica Ceci, Riccardo Turchi, Savina Apolloni, Martina Milani, Ilaria Della Valle, Marta Tiberi, Valerio Chiurchiù, Nadia D'Ambrosi, Silvia Pedretti, Nico Mitro, Cinzia Volontè, Susanna Amadio, Katia Aquilano, and Daniele Lettieri Barbato

DOI: <https://doi.org/10.26508/lsa.202402609>

Corresponding author(s): *Daniele Lettieri Barbato, University of Rome Tor Vergata and Katia Aquilano, University of Rome Tor Vergata*

Review Timeline:

Submission Date:	2024-01-22
Editorial Decision:	2024-03-11
Revision Received:	2024-03-22
Editorial Decision:	2024-04-02
Revision Received:	2024-04-05
Accepted:	2024-04-08

Transaction Report:

March 11, 2024

Re: Life Science Alliance manuscript #LSA-2024-02609-T

Daniele Lettieri Barbato
University of Rome Tor Vergata
Italy

Dear Dr. Lettieri Barbato,

Thank you for submitting your manuscript entitled "Frataxin Deficiency-Induced Metabolic Shift Promotes Microglial Inflammation" to Life Science Alliance. The manuscript was assessed by expert reviewers, whose comments are appended to this letter. We invite you to submit a revised manuscript addressing the Reviewer comments.

Thank you for this interesting contribution to Life Science Alliance. We are looking forward to receiving your revised manuscript.

Sincerely,

B. MANUSCRIPT ORGANIZATION AND FORMATTING:

Reviewer #1 (Comments to the Authors (Required)):

1. In the present manuscript Sciarretta and collaborators, using single-cell RNA sequencing, identify an inflammatory microglial response in the cerebellum from a Friedreich Ataxia (FRDA) mouse model. They also perform a metabolomic analyses which reveals increased itaconate production in these cells. Moreover, they show that butyrate treatment counteracts these immunometabolic changes. The authors conclude that their findings shed light on how cerebellar microglia activation contributes to Friedreich Ataxia. They also conclude that their findings suggest that butyrate could be a readily accessible, safe therapeutic option for alleviating neurological symptoms in this disease.

Overall, the conclusions are of interest for understanding FRDA disease mechanism, and the results obtained with butyrate could contribute to find a cure for this disease.

2. Nevertheless, the manuscript has several weaknesses that should be addressed:

- In materials and methods, it is indicated that all metabolites analyzed were previously validated by pure standards.

Nevertheless, given the relevance of itaconate identification for the conclusions raised, more detailed data about identification of this compound by LC-MS should be provided: m/z, retention time, standard used (reference), and also indicate if MS/MS spectra was performed (and in that case provide it as a supplemental figure).

- Only one shRNA was used in the experiments performed with BV2 cells. As shRNA can lead to off-target effects, it is common to validate results with additional shRNA sequences. This is a limitation of this study that should either be corrected or acknowledged in the discussion. Besides, more detailed information about the shRNA sequence(s) used should be provided in the methods section.

- All generated RNAseq data should be deposited into public repositories, and access information indicated in the manuscript.

- According to text, lines 110- 111 "GO terms for biological processes of the top 200 down-regulated genes revealed a reduced mitochondrial oxidative capacity in microglia/macrophages of KIKO mice (Fig. 2D)". Nevertheless, In figure 2D, fatty acids biosynthesis and other lipid related processes are indicated. Please, clarify this point. Figure 3B indicates changes in citrate , α KG, OAA between SCR and FXN- cells. Why text indicates that there are no changes? (lines 128-129). Please clarify this point.

- Quantitative data is missing for western blot experiments shown in Fig 5D and J. Please, provide these data and indicate how many replicates were performed.

- Indicate reference and provider from the antibodies used (in methods section).

3. Additional issues.

Authors should discuss several issues related to itaconate production and TCA cycle alterations: i) It has been shown by other authors that itaconate is produced in macrophages after downregulation of isocitrate dehydrogenase (IDH). In that case, citrate and isocitrate accumulate and carbon flux is redirected to the production of itaconate, while production of α -KG (IDH product) is decreased. This model does not fit with the results presented in this manuscript, as both α -KG and itaconate are increased in KIKO mice, while citrate is decreased. Please, discuss this point; ii) Besides, itaconate production requires aconitase activity, which is an iron-sulfur enzyme, and which may be affected by frataxin deficiency. How itaconate production may be increased in a condition where aconitase activity may be decreased?

Reviewer #2 (Comments to the Authors (Required)):

In this manuscript, Sciarretta et al. study the impact of frataxin on the cerebellum using a knockout mouse model. Through scRNA-seq assay, they identified microglia as being highly impacted by the loss of expression of FXN and suggested that butyrate treatment could restore metabolic deficiency, inflammation, and motor phenotype of these animals. While identifying which cell type of driving disease is extremely important, this study lacks controls and validation to support the authors' conclusions.

A) Single-cell isolation of the mouse cerebellum is a complex process that can lead to artificial signatures and aberrant responses of the cells or a change in cell proportion. Few details are given about the cell isolation in the Materials and Methods section (how were the cells isolated? What are the markers used to sort the cells?), no quality control measures have been shown (UMAP per animal, number of UMI, level of mitochondrial genes...), and not enough details are given about the analysis (package version, code availability). This is crucial as it is the base of this manuscript. Moreover, comparing the cell type obtained with previously published studies such as PMID: 34616064 is essential. Many circulating immune cells are identified in this dataset; it is necessary to know if these cells are infiltrating or were carried over from poor perfusion.

B) The authors claim that microglia is the most changing cell type in their dataset and have done follow-up experiments using primary microglia. Additional markers are needed to confirm the cells identified in their single-cell dataset are microglia and not circulating or border-associated macrophages (see PMID: 37232741 for examples). In vivo validation using in situ hybridization or immunohistochemistry is needed to confirm DEG from Fig 2C. Also, based on the details given in the materials and methods section, primary microglia are cultured without CSF1, which is unusual as microglia survival depends on CSF1 (for a standard protocol for primary microglia culture PMID: 30414379). Immunocytochemistry is needed to confirm microglia are the cells grown in vitro. Finally, as recommended by PMID: 36327895, avoid using M1 and M2 to describe microglia signature

C) What is the rationale for the time points chosen for the animal experiments? The age of onset in Friedrich ataxia is between 5 and 15 years old, but all experiments using mice were done in 6-8-month-old animals.

D) Finally, this manuscript needs to be rewritten in a more detailed and accurate way. For example, Line 118 - are the BV2 cells a stable cell line or infected at every differentiation? In either case, more details are needed in the materials and methods section

line 153 How glucose was measured in Fig 4 should be explained in the text

Multiple abbreviations are missing line 162 BMDM and line 161 BUT, among others

Figure 2C- If these graphs are based on RNAseq values, the y-axis should be gene expression, not protein

Cell are infected with lentivirus not transfected

RESPONSE TO REVIEW

Reviewer #1 (Comments to the Authors (Required)):

1. In the present manuscript Sciarretta and collaborators, using single-cell RNA sequencing, identify an inflammatory microglial response in the cerebellum from a Friedreich Ataxia (FRDA) mouse model. They also perform a metabolomic analyses which reveals increased itaconate production in these cells. Moreover, they show that butyrate treatment counteracts these immunometabolic changes. The authors conclude that their findings shed light on how cerebellar microglia activation contributes to Friedreich Ataxia. They also conclude that their findings suggest that butyrate could be a readily accessible, safe therapeutic option for alleviating neurological symptoms in this disease. Overall, the conclusions are of interest for understanding FRDA disease mechanism, and the results obtained with butyrate could contribute to find a cure for this disease.

2. Nevertheless, the manuscript has several weaknesses that should be addressed:

- In materials and methods, it is indicated that all metabolites analyzed were previously validated by pure standards. Nevertheless, given the relevance of itaconate identification for the conclusions raised, more detailed data about identification of this compound by LC-MS should be provided: m/z, retention time, standard used (reference), and also indicate if MS/MS spectra was performed (and in that case provide it as a supplemental figure).

Reply: We thank the reviewer for his/her insightful comment. Below we provide the information required for itaconate:

m/z	128.7
Ion Source	ESI negative
Instrument	API3500, Triple Quad MS
Retention Time	0.84 min
Multiple reaction monitoring	129/85
Standard used for reference	Mass spectrometry Metabolites Library (Sigma Aldrich) catalog number I29204

The MS/MS spectra of itaconic acid are shown below as well as in the **new Figure S2B**.

Itaconic acid
Below are the parameters set on the instrument for the MS/MS spectrum of itaconic acid.

- Declustering potential: -15.91 V;
- Collision energy: -13.14 V;
- Collision Cell Exit Potential: -7.46 V;
- Entrance potential: -4.72 V.

- Only one shRNA was used in the experiments performed with BV2 cells. As shRNA can lead to off-target effects, it is common to validate results with additional shRNA sequences. This is a limitation of this study that should either be corrected or acknowledged in the discussion.

Reply. I appreciate the reviewer's insightful critique regarding the utilization of a single shRNA sequence in our study. Acknowledging the importance of this concern, we have duly incorporated the reviewer's suggestion. In the revised

manuscript, we have explicitly addressed this limitation by discussing the absence of further validation with an alternative sequence (see new section "limitation of the study", lines 281-284).

Besides, more detailed information about the shRNA sequence(s) used should be provided in the methods section.

Reply. I would like to express my gratitude to Reviewer #1 for the valuable suggestion. We have taken your feedback into consideration by providing more details about the shRNAs used in the methods section. You can find the updated details in the revised version of the manuscript, specifically in the methods section (lines 359-360).

- All generated RNAseq data should be deposited into public repositories, and access information indicated in the manuscript.

Reply. I am grateful to the reviewer for bringing this to my attention. In response, we have taken the necessary steps to ensure the accessibility of our RNAseq data by depositing it on GEO datasets (Master Report: GSE261655; Bulk RNAseq: GSE261653; scRNAseq: GSE261654).

- According to text, lines 110- 111 "GO terms for biological processes of the top 200 down-regulated genes revealed a reduced mitochondrial oxidative capacity in microglia/macrophages of KIKO mice (Fig. 2D)". Nevertheless, In figure 2D, fatty acids biosynthesis and other lipid related processes are indicated. Please, clarify this point.

Reply. I sincerely apologize to Reviewer #1 for the oversight that resulted in the lack of clarity in the functional enrichment analysis of up- and downregulated genes in microglia isolated from the cerebellum of kiko mice. We have rectified this error by implementing the modifications suggested by Reviewer #1. Specifically, we have now provided a clear indication of the outcomes reported by the enrichment analysis, thus enhancing the comprehensibility and accuracy of our findings (see lines 117-118).

Figure 3B indicates changes in citrate, aKG, OAA between SCR and FXN- cells. Why text indicates that there are no changes? (lines 128-129). Please clarify this point.

Reply. I extend my gratitude to Reviewer #1 for highlighting the error, and to provide a clearer explanation, we have revised the description of our metabolomics results with greater precision. In the updated version of the manuscript, we have specifically addressed the changes observed in metabolite levels with more accuracy. Notably, we have elucidated that while LPS treatment effectively increased the levels of citrate, aKG, and OAA in scr microglia, the absence of frataxin resulted in a reduction of citrate and OAA levels, while aKG levels exhibited an increased level (see lines 141-142).

- Quantitative data is missing for western blot experiments shown in Fig 5D and J. Please, provide these data and indicate how many replicates were performed.

Reply. I thank Reviewer#1 and in the new version of the manuscript the densitometry and number of replicates for the immunoblots have been reported (see new Fig. 5D and 5J).

- Indicate reference and provider from the antibodies used (in methods section).

Reply. Ok, reference and provider from the antibodies has been included in the new version of methods section (see lines 551-552).

3. Additional issues.

Authors should discuss several issues related to itaconate production and TCA cycle alterations: i) It has been shown by other authors that itaconate is produced in macrophages after downregulation of isocitrate dehydrogenase (IDH). In that case, citrate and isocitrate accumulate and carbon flux is redirected to the production of itaconate, while production of α -KG (IDH product) is decreased. This model does not fit with the results presented in this manuscript, as both α -KG and itaconate are increased in KIKO mice, while citrate is decreased. Please, discuss this point; ii) Besides, itaconate production requires aconitase activity, which is an iron-sulfur enzyme, and which may be affected by frataxin deficiency. How itaconate production may be increased in a condition where aconitase activity may be decreased?

Reply. I sincerely appreciate the reviewer's insightful comment regarding the need for clarification on this crucial aspect of our research. We concur with the reviewer's observation regarding the challenge in deciphering the precise metabolic pathways leading to itaconate production. While our study did not include an analysis of metabolic fluxes, the observed elevation of aKG and itaconate in cells lacking FXN suggests a potential link between increased aKG levels and itaconate accumulation. Furthermore, supporting evidence from previous studies suggests that the incubation of cells with glutamine should yield itaconate as the primary isotopologue in the first turn of the TCA cycle. This pathway involves glutamine hydrolysis to glutamate, followed by transamination and the production of α -ketoglutarate, which subsequently enters the TCA cycle. Ultimately, this leads to the formation of cis-aconitate, which is then site-specifically decarboxylated to itaconate (Puchalska et al., 2018; Strelko et al., 2011). However, it is important to acknowledge that our study does not conclusively rule out potential limitations in the metabolic flux of itaconate removal in FXN-deficient microglia.

Specifically, we did not quantify the levels of metabolites derived from with itaconate, such as mesaconate and itaconyl-CoA, nor did we evaluate the expression levels of enzymes responsible for itaconate clearance (Marcero et al., 2021).

Recognizing the significance of this point raised by the reviewer, we have decided to incorporate these limitations into our manuscript (see lines 282-292).

We also recognize that aconitase, being crucial for the conversion of citrate to isocitrate in the TCA cycle, plays an essential role in the metabolic pathway leading to itaconate production. Despite this, our findings indicate an increase in itaconate production in the KIKO mouse model. We propose several hypotheses that could explain this apparent paradox. It is possible that residual aconitase activity, albeit reduced, is sufficient to support a certain level of itaconate production. The severity of frataxin deficiency's impact on aconitase could vary, allowing for some degree of itaconate

synthesis to occur. Alternatively, given the role of itaconate as an immunometabolite with anti-inflammatory properties, its increased production might reflect a cellular response to mitochondrial dysfunction and oxidative stress. This adaptive response could promote the upregulation of itaconate synthesis as a protective mechanism (lines 260-269 in Discussion).

By doing so, we aim to provide a more comprehensive understanding of the complexities surrounding itaconate metabolism in FXN-deficient microglia.

Reviewer #2 (Comments to the Authors (Required)):

In this manuscript, Sciarretta et al. study the impact of frataxin on the cerebellum using a knockout mouse model. Through scRNA-seq assay, they identified microglia as being highly impacted by the loss of expression of FXN and suggested that butyrate treatment could restore metabolic deficiency, inflammation, and motor phenotype of these animals. While identifying which cell type of driving disease is extremely important, this study lacks controls and validation to support the authors' conclusions.

A) Single-cell isolation of the mouse cerebellum is a complex process that can lead to artificial signatures and aberrant responses of the cells or a change in cell proportion. Few details are given about the cell isolation in the Materials and Methods section (how were the cells isolated? What are the markers used to sort the cells?)...

Reply: We fully acknowledge and appreciate the constructive criticism provided by the reviewer. In response to his/her feedback, we have made significant revisions to the methods section. Specifically, we have incorporated a dedicated section in the Methods that thoroughly addresses the points highlighted by the reviewer (see new section "Generation of cerebellar single-cell suspensions", lines 369-383)

...no quality control measures have been shown (UMAP per animal, number of UMI, level of mitochondrial genes...), and not enough details are given about the analysis (package version, code availability). This is crucial as it is the base of this manuscript.

Reply: I express my gratitude to the reviewer for their valuable observation, which has proven instrumental in enhancing the quality of our work. In response to their feedback, we have enriched the Method section with additional details regarding the scRNAseq data processing (see lines 395-400). In particular, raw reads were aligned to the *Mus musculus* (mm10) reference genome using CellRanger v7.1.0. Next the R package "Seurat" (v4.4.0) was used for the downstream analysis. KIKO sample: the raw data returned 32285 genes (features) and 3353 cells (barcodes), after the quality control and filtering steps were retained only cells with more than 500 features, gene counts greater than 1000, and mitochondrial content less than 10% (see below confidential figure). Outliers, defined as cells with more than 10000 features and counts exceeding 20000, were removed, resulting in a final subset composed of 2358 cells. WT sample: 2854 cells were filtered by the same approach used for the KIKO sample, resulting in a final WT subset composed of 2442 cells.

...Moreover, comparing the cell type obtained with previously published studies such as PMID: 34616064 is essential. Many circulating immune cells are identified in this dataset; it is necessary to know if these cells are infiltrating or were carried over from poor perfusion.

Reply: I value the insightful suggestion provided by the reviewer regarding the comparison of our scRNAseq dataset with others documented in the literature. We find the prospect of such a comparison to be quite intriguing. However, in order to address potential biases stemming from variations in the age of the mice, we have opted to conduct the comparative analysis with datasets that encompass mice of different ages from ours. Specifically, we have chosen to refrain from comparing our dataset with others that involve mice at various developmental stages (At E18, P0, P4, P8, P12, P16, and P60)(Kozareva et al., 2021). In our study, we concentrated on mice at an advanced age (P180), for which we did not find scRNAseq data available for the cerebellum in the existing scientific literature. Furthermore, our results refer to the first scRNAseq dataset in the literature in which the transcriptomic profile of the cerebellum of symptomatic KIKO mice (P180) was analyzed at the single-cell level. Thank you for bringing this to our attention, and we remain fully committed to addressing any further questions or concerns you may have.

B) The authors claim that microglia is the most changing cell type in their dataset and have done follow-up experiments using primary microglia. Additional markers are needed to confirm the cells identified in their single-cell dataset are microglia and not circulating or border-associated macrophages (see PMID: 37232741 for examples).

Reply: I express my gratitude to the reviewer for their insightful observation. In alignment with their suggestion, and in concurrence with other authors, we recognize the complexity inherent in discerning between microglia and infiltrated macrophages solely through gene expression analysis. Indeed, both cell types share a considerable number of genes, such as *Ptpcr*, *Adgre1*, and *Ilgam*. However, it is noteworthy that there exist specific markers exclusive to microglia, aiding in their distinct identification. Notable among these markers are the purinergic receptor P2Y12 (*P2ry12*), the hexosaminidase subunit beta (*Hexb*), the siglech sialic acid binding Ig-like lectin H (*Siglech*), the transmembrane protein 119 (*Tmem119*), and the Spalt-like transcription factor 1 (*Sall1*). We agree that incorporating the expression of these markers with commonly-used microglia-specific genes, such as *Csf1r*, *Cd74*, *C1qa*, *Ptgs1*, *Gpr34*, and *Lag3*, to bolster the comprehensive characterization of microglial populations in our study (see new Fig. S1A). Moreover, we have taken significant steps to address the specificity of our findings regarding microglia. In instances where we employed non-highly specific markers like CD11b+, we have consciously tempered our assertions about microglia. Specifically, in sections where we isolated CD45+/CD11+ cells and in the immunohistochemistry images (new Fig. S1B), we have opted to use the term "macrophage/microglia" to accurately reflect the potential involvement of both cell types. This adjustment ensures a more nuanced interpretation of our results and enhances the accuracy of our findings.

...In vivo validation using in situ hybridization or immunohistochemistry is needed to confirm DEG from Fig 2C.

Reply: I wholeheartedly agree with the reviewer's suggestion to complement the evidence from RNAseq and scRNAseq with other methodologies. To describe the morphological features of CD11b+ cells in the cerebellum of KIKO mice, our aim was to employ Sholl analysis, a method that allows for the examination of both microglia and neural arborizations. (Ferreira et al., 2014; Ristanovic et al., 2006). Morphological analysis revealed that CD11b+ cells in KIKO mice exhibited longer ramification lengths compared to WT mice (new Fig. S1C, upper panel), suggesting an increased reactivity of microglia to inflammatory stimuli (Vidal-Itriago et al., 2022; Ziebell et al., 2015). Moreover, to assess mitochondrial functionality (related to top 200 down-regulated genes), CD45+/CD11b+ cells were isolated from the cerebellum, and mitochondrial membrane potential ($\Delta\Psi$ M) was measured. As expected, a reduced $\Delta\Psi$ M was detected in cerebellar macrophage/microglia of KIKO mice compared to WT mice (new Fig. S1D, lower panel).

...Also, based on the details given in the materials and methods section, primary microglia are cultured without CSF1, which is unusual as microglia survival depends on CSF1 (for a standard protocol for primary microglia culture PMID: 30414379). Immunocytochemistry is needed to confirm microglia are the cells grown *in vitro*.

Reply: I wish to extend my sincere appreciation for your insightful suggestion. Concerning your inquiry about the use of CSF1 in *in vitro* cultures of microglia, I fully recognize the importance of clarity and rigor in experimental procedures. Allow me to underscore that the protocol employed in our study has been meticulously developed and validated within our lab. Specifically, the methodology employed in this paper closely adheres to the approach outlined by Saura et al. (2007), with minor adaptations as elucidated by Apolloni et al. (J Immunology, 2013; Molecular Neurobiology, 2016). In this protocol, microglia are cultured in conditioned medium derived from mixed glia cultures, ensuring the presence of all necessary trophic factors. As delineated by Saura, the addition of CSF serves as an activating agent, promoting microglial proliferation compared to untreated cells. Therefore, for experiments under basal conditions, the addition of CSF is deemed unnecessary.

Furthermore, to enhance transparency and provide assurance regarding the quality of our microglia cultures, we have incorporated immunofluorescent images verifying their purity *in vitro* (see new Figure S2B).

Representative fluorescence images of KIKO cerebellar microglia in culture stained with anti-Cd11b (red). Nuclei are marker with DAPI (blue). Scale bar, 20 µm

Finally, as recommended by PMID: 36327895, avoid using M1 and M2 to describe microglia signature

Reply: We wholeheartedly agree with the reviewer and greatly appreciate their suggestion to reassess the definition of microglia, such as the distinction between M1 and M2 phenotypes, in light of the referenced article. In the revised version of the manuscript, we have incorporated the following terms: "reactive to" instead of activated and "loss in homeostatic function" instead of M1 (see abstract; line 90; line 168-169; line 226; line 227; line 264).

C)What is the rationale for the time points chosen for the animal experiments? The age of onset in Friedrich ataxia is between 5 and 15 years old, but all experiments using mice were done in 6-8-month-old animals.

Reply. I concur with the observation made by Reviewer #2 regarding the need for clarification on this significant point. Characterization of KIKO mice performed at The Jackson Laboratory revealed that starting at 6 months of age, these animals exhibit an abnormal "weaving" gait when subjected to a forced treadmill walk. This phenotype occurs with increasing penetrance as the mice age. This is consistent with several works demonstrating that KIKO mice show metabolic alterations starting from 6-8 months of age (McMackin et al., 2017; Turchi et al., 2020).

D) Finally, this manuscript needs to be rewritten in a more detailed and accurate way. For example, Line 118 - are the BV2 cells a stable cell line or infected at every differentiation?

Reply. I am grateful to the reviewer for the valuable suggestion. In the revised version of our manuscript, we have specified that the BV2 cell lines with downregulated FXN were stabilized (see lines 131-136).

In either case, more details are needed in the materials and methods section line 153 How glucose was measured in Fig 4 should be explained in the text Multiple abbreviations are missing line 162 BMDM and line 161 BUT.

Reply. I appreciate the valuable suggestion provided by the reviewer. In the revised version of our manuscript, we have made improvements by providing a clearer explanation of how glucose was measured (lines 496-498) and by correcting several abbreviations (lines 167-220).

...among others Figure 2C- If these graphs are based on RNAseq values, the y-axis should be gene expression, not protein

Reply. I acknowledge the reviewer's suggestion regarding modifying the y bar of the RNAseq figure, and I regret any inconvenience caused by the inability to make this adjustment. The graph was generated using Funrich3.0, a tool that provides functional enrichment analysis information of genes, using the term "proteins."

...Cell are infected with lentivirus not transfected.

Reply. I apologize to the reviewer for this oversight. In the updated version of the text, we have implemented the suggested change.

REFERENCES

- Ferreira, T.A., Blackman, A.V., Oyrer, J., Jayabal, S., Chung, A.J., Watt, A.J., Sjostrom, P.J., and van Meyel, D.J. (2014). Neuronal morphometry directly from bitmap images. *Nat Methods* 11, 982-984. 10.1038/nmeth.3125.
- Kozareva, V., Martin, C., Osorno, T., Rudolph, S., Guo, C., Vanderburg, C., Nadaf, N., Regev, A., Regehr, W.G., and Macosko, E. (2021). A transcriptomic atlas of mouse cerebellar cortex comprehensively defines cell types. *Nature* 598, 214-219. 10.1038/s41586-021-03220-z.
- Marcero, J.R., Cox, J.E., Bergonia, H.A., Medlock, A.E., Phillips, J.D., and Dailey, H.A. (2021). The immunometabolite itaconate inhibits heme synthesis and remodels cellular metabolism in erythroid precursors. *Blood Adv* 5, 4831-4841. 10.1182/bloodadvances.2021004750.
- McMackin, M.Z., Henderson, C.K., and Cortopassi, G.A. (2017). Neurobehavioral deficits in the KIKO mouse model of Friedreich's ataxia. *Behav Brain Res* 316, 183-188. 10.1016/j.bbr.2016.08.053.
- Puchalska, P., Huang, X., Martin, S.E., Han, X., Patti, G.J., and Crawford, P.A. (2018). Isotope Tracing Untargeted Metabolomics Reveals Macrophage Polarization-State-Specific Metabolic Coordination across Intracellular Compartments. *iScience* 9, 298-313. 10.1016/j.isci.2018.10.029.
- Ristanovic, D., Milosevic, N.T., and Stulic, V. (2006). Application of modified Sholl analysis to neuronal dendritic arborization of the cat spinal cord. *J Neurosci Methods* 158, 212-218. 10.1016/j.jneumeth.2006.05.030.
- Strelko, C.L., Lu, W., Dufort, F.J., Seyfried, T.N., Chiles, T.C., Rabinowitz, J.D., and Roberts, M.F. (2011). Itaconic acid is a mammalian metabolite induced during macrophage activation. *J Am Chem Soc* 133, 16386-16389. 10.1021/ja2070889.
- Turchi, R., Tortolici, F., Guidobaldi, G., Iacovelli, F., Falconi, M., Rufini, S., Faraonio, R., Casagrande, V., Federici, M., De Angelis, L., et al. (2020). Frataxin deficiency induces lipid accumulation and affects thermogenesis in brown adipose tissue. *Cell Death Dis* 11, 51. 10.1038/s41419-020-2253-2.
- Vidal-Itriago, A., Radford, R.A.W., Aramideh, J.A., Maurel, C., Scherer, N.M., Don, E.K., Lee, A., Chung, R.S., Graeber, M.B., and Morsch, M. (2022). Microglia morphophysiological diversity and its implications for the CNS. *Front Immunol* 13, 997786. 10.3389/fimmu.2022.997786.

Ziebell, J.M., Adelson, P.D., and Lifshitz, J. (2015). Microglia: dismantling and rebuilding circuits after acute neurological injury. *Metab Brain Dis* 30, 393-400. 10.1007/s11011-014-9539-y.

April 2, 2024

RE: Life Science Alliance Manuscript #LSA-2024-02609-TR

Dr. Daniele Lettieri Barbato
University of Rome Tor Vergata
Via della Ricerca Scientifica 1
Rome 00133
Italy

Dear Dr. Lettieri Barbato,

Thank you for submitting your revised manuscript entitled "Frataxin Deficiency Drives a Shift from Mitochondrial to Glucose Metabolism in Microglia". We would be happy to publish your paper in Life Science Alliance pending final revisions necessary to meet our formatting guidelines.

- please be sure that the authorship listing and order is correct, and that they match in the system and manuscript file
- please upload all figure files as individual ones, including the supplementary figure files; all figure legends should only appear in the main manuscript file
- please add your main and supplementary figure legends to the main manuscript text after the references section
- please add ORCID ID for the secondary corresponding author -- they should have received instructions on how to do so
- please add a Summary Blurb/Alternate Abstract to our system
- please add the Twitter handle of your host institute/organization as well as your own or/and one of the authors in our system
- please note that the titles in the system and on the manuscript text must match
- please use the [10 author names et al.] format in your references (i.e., limit the author names to the first 10)
- please add an Author Contributions section to your main manuscript text

FIGURE CHECKS:

- please add sizes next to the blots in Figures 3 and 5

A. FINAL FILES:

B. MANUSCRIPT ORGANIZATION AND FORMATTING:

Sincerely,

Reviewer #1 (Comments to the Authors (Required)):

The authors have addressed the concerns raised in the previous review.

I suggest increasing the size from the m/z values (fragments and precursor) in supplemental figure 2B. I'm afraid that in the current size they might not be visible in the final version

April 8, 2024

RE: Life Science Alliance Manuscript #LSA-2024-02609-TRR

Dr. Daniele Lettieri Barbato
University of Rome Tor Vergata
Via della Ricerca Scientifica 1
Rome 00133
Italy

Dear Dr. Lettieri Barbato,

Thank you for submitting your Research Article entitled "Fratxin Deficiency Shifts Metabolism to Promote Reactive Microglia via Glucose Catabolism". It is a pleasure to let you know that your manuscript is now accepted for publication in Life Science Alliance. Congratulations on this interesting work.

DISTRIBUTION OF MATERIALS:

Again, congratulations on a very nice paper. I hope you found the review process to be constructive and are pleased with how the manuscript was handled editorially. We look forward to future exciting submissions from your lab.

Sincerely,
